# Centering and symmetry breaking in confined contracting actomyosin networks

Niv Ierushalmi[1], Maya Malik-Garbi[1], Angelika Manhart[2], Enas Abu Shah[1,3], Bruce L Goode[4], Alex Mogilner[5], Kinneret Keren[1,6]*

[1]Department of Physics, Technion- Israel Institute of Technology, Haifa, Israel; [2]Department of Mathematics, University College London, London, United Kingdom; [3]Kennedy Institute of Rheumatology, University of Oxford, Oxford, United Kingdom; [4]Department of Biology, Brandeis University, Waltham, United States; [5]Courant Institute of Mathematical Sciences and Department of Biology, New York University, New York, United States; [6]Network Biology Research Laboratories and Russell Berrie Nanotechnology Institute, Technion – Israel Institute of Technology, Haifa, Israel

**Abstract** Centering and decentering of cellular components is essential for internal organization of cells and their ability to perform basic cellular functions such as division and motility. How cells achieve proper localization of their organelles is still not well-understood, especially in large cells such as oocytes. Here, we study actin-based positioning mechanisms in artificial cells with persistently contracting actomyosin networks, generated by encapsulating cytoplasmic *Xenopus* egg extracts into cell-sized 'water-in-oil' droplets. We observe size-dependent localization of the contraction center, with a symmetric configuration in larger cells and a polar one in smaller cells. Centering is achieved via a hydrodynamic mechanism based on Darcy friction between the contracting network and the surrounding cytoplasm. During symmetry breaking, transient attachments to the cell boundary drive the contraction center to a polar location. The centering mechanism is cell-cycle dependent and weakens considerably during interphase. Our findings demonstrate a robust, yet tunable, mechanism for subcellular localization.

*For correspondence:
kinneret@technion.ac.il

Competing interests: The authors declare that no competing interests exist.

## Introduction

The proper localization of cellular components is essential for a variety of cell functions (*Rafelski and Marshall, 2008*). Depending on the cellular context, some components need to be positioned at the center of the cell, whereas other components must assume a polar, decentered localization (*van Bergeijk et al., 2016*). For example, depending on the cell type and its stage in the cell cycle, the cell nucleus has to be positioned at the cell center or asymmetrically localized, whereas abnormal nuclear positioning can lead to various pathologies and disease (*Gundersen and Worman, 2013*). Similarly, the centrosome is located at the geometrical center of cells under many conditions, yet in other cases it is required at the cell periphery where it serves as the base of the primary cilium (*Letort et al., 2016*). Cells use diverse mechanisms to control their internal organization and dynamically regulate the positioning of their organelles in response to various internal and external cues. While much research has been devoted to the study of the various cellular positioning mechanisms, many basic questions remain open. This is particularly true in large cells such as oocytes (*Mitchison, 2012*), where movements must be coordinated over large scales and it is often still unclear what drives movement in the first place and how the proper localization is stabilized.

**eLife digest** In order to survive, cells need to react to their environment and change their shape or the localization of their internal components. For example, the nucleus – the compartment that contains the genetic information – is often localized at the center of the cell, but it can also be positioned at the side, for instance when cells move or divide asymmetrically.

Cells use multiple positioning mechanisms to move their internal components, including a process that relies on networks of filaments made of a protein known as actin. These networks are constantly remodeled as actin proteins are added and removed from the network. Embedded molecular motors can cause the network of actin filaments to contract and push or pull on the compartments. Yet, the exact way these networks localize components in the cell remains unclear, especially in eggs and other large cells.

To investigate this question, Ierushalmi et al. studied the actin networks in artificial cells that they created by enclosing the contents of frog eggs in small droplets surrounded by oil. This showed that the networks contracted either to the center of the cell or to its side. Friction between the contracting actin network and the fluid in the cell generated a force that tends to push the contraction center towards the middle of the cell. In larger cells, this led to the centering of the actin network. In smaller cells however, the network transiently attached to the boundary of the cell, leading the contraction center to be pulled to one side.

By developing simpler artificial cells that mimic the positioning processes seen in real-life cells, Ierushalmi et al. discovered new mechanisms for how cells may center or de-center their components. This knowledge may be useful to understand diseases that can emerge when the nucleus or other compartments fail to move to the right location, and which are associated with certain organs developing incorrectly.

Transport of cellular components depends on biophysical processes that rely on the cell's cytoskeleton. The mechanisms involved are diverse, typically employing either microtubules, the actin cytoskeleton, or both, together with their respective molecular motors (*Gundersen and Worman, 2013*; *Mitchison, 2012*; *Mullins, 2010*; *Xie and Minc, 2020*). The mechanisms also vary in terms of the relevant length scales at which they operate, as the physical requirements for positioning objects in a small cell are different from those in an extremely large oocyte (*Mitchison, 2012*; *Wühr et al., 2009*; *Mogessie et al., 2018*). Many positioning mechanisms rely on microtubule asters radiating from the organelle being positioned, impinging on the cortex at the cell boundary and generating pushing and/or pulling forces against it (*Wühr et al., 2009*). For example, the spindle in dividing cells is often centered by astral microtubules that emanate radially from the centrosomes and interact with dynein motors at the cortex (*Grill and Hyman, 2005*). However, this mechanism depends on the availability of long enough microtubules that can directly interact with the cell boundary. In large cells such as oocytes, this is often not the case and various actin-based localization schemes are found. These include, for example, nuclear centering in mouse oocytes which depends on gradients in actin-dependent active diffusion (*Almonacid et al., 2015*), chromosome congression in starfish oocytes where a contracting actin network carries the chromosomes like a fishnet (*Lénárt et al., 2005*), or ooplasm segregation in zebrafish oocytes where differential friction with a contracting actin network drags the ooplasm toward the animal pole (*Shamipour et al., 2019*; *Ierushalmi and Keren, 2019*).

The study of cellular localization mechanisms has greatly benefitted from in vitro work on cell-free systems, that make it possible to study the localization schemes in a simplified, well-controlled environment, and isolate the basic biophysical and biochemical processes involved (*Mullins, 2010*; *Holy et al., 1997*; *Laan et al., 2012*; *Abu Shah and Keren, 2014*). A notable example is the work on localization of microtubule asters in micro-fabricated, cell-sized compartments. Early work showed that microtubule assembly and disassembly dynamics are sufficient for centering of microtubules asters (*Holy et al., 1997*), while more recently the influence of cortex-bound microtubule motors was studied by attaching motors to the compartment's interface (*Laan et al., 2012*). These experiments, together with theoretical modeling (*Grill et al., 2001*; *Vogel et al., 2009*), showed

how the interaction between the tips of microtubules and the cell boundary generates pushing and pulling forces that lead to robust centering under a variety of conditions.

Localization in large cells where the cytoskeletal elements do not span the entire system, cannot rely on direct interaction between the cytoskeleton and the cell boundary. Rather, the centering mechanisms must involve indirect sensing of the cell boundary, to be able to define the cell center without directly interacting with it (*Wühr et al., 2009*). Here we use a recently developed in vitro system that self organizes to form persistently contracting bulk actin networks within cell-sized compartments (*Malik-Garbi et al., 2019*), to demonstrate a hydrodynamic centering mechanism that can function in very large cells in the absence of any direct interaction between the cytoskeleton and the cell boundary.

Our system is based on encapsulation of *Xenopus* egg extracts in cell-sized water-in-oil emulsions (*Abu Shah and Keren, 2014*; *Malik-Garbi et al., 2019*; *Pinot et al., 2012*; *Tang et al., 2018*). The system self-organizes to form persistently contracting actomyosin networks surrounding an aggregate that forms around the contraction center (*Malik-Garbi et al., 2019*). We observe size-dependent localization of the aggregate: large droplets are symmetric with the aggregate positioned at the center, whereas smaller droplets are polar with the aggregate near the boundary. The centering and decentering of the contraction center resemble cellular centering and decentering as seen for example during nuclear centering and spindle migration in mammalian oocytes (*Almonacid et al., 2018*; *Uraji et al., 2018*) and plant eggs (*Ohnishi and Okamoto, 2017*), and can serve as a simplified model to study actin-based localization in large cells.

We show that the centered state is stable against large perturbations and suggest a hydrodynamic active centering mechanism that is based on an imbalance of the Darcy friction forces between the contracting actomyosin network and the cell's cytoplasm. We use mathematical modeling to show how the displacement of the contraction center from the center of the droplet is translated into an asymmetry in the actin network density, and how this in turn leads to an effective centering force with spring-like properties. We further show that the model correctly predicts how the network properties affect the centering dynamics under various conditions, including different cell-cycle states and biochemical conditions. The size-dependent localization of the contraction center arises from a competition between the hydrodynamic centering force, and a decentering force due to engagement between the contracting network and the boundary, which is more prominent in smaller droplets. Finally, we discuss the implication of these findings for intracellular centering and symmetry breaking, and suggest future experiments to examine if the proposed mechanisms are at play in cellular processes.

## Results

### Size-dependent localization of the contraction center in artificial cells

Persistently contracting actin networks are generated in cell-like compartments by encapsulating cytoplasmic *Xenopus* egg extract in water-in-oil emulsion droplets (*Abu Shah and Keren, 2014*; *Malik-Garbi et al., 2019*; *Pinot et al., 2012*; *Tang et al., 2018*). Endogenous actin nucleation activities induce the formation of bulk actin networks, which undergo continuous myosin-driven contraction (*Malik-Garbi et al., 2019*). A dense 'exclusion zone' forms around the contraction center within minutes, as the network contracts and accumulates particulates from the (crude) extract into a dense aggregate. We find that the droplets are typically in one of two configurations: a symmetric state or a polar state (*Figure 1*). In the symmetric state, the aggregate is localized near the middle of the droplet and the network exhibits spherically symmetric density and flow patterns (*Figure 1a–c*, *Video 1*). In the polar state, the aggregate is positioned near the droplet's boundary and the network displays a flow pattern that is skewed toward the side (*Figure 1a–c*, *Video 2*).

The observed configurations, both the symmetric one and the polar one, reflect a dynamic steady-state in which the system self-organizes into persistent contractile flow patterns, which remain nearly stationary over time scales that are considerably longer than the characteristic time scale for network contraction and turnover (~1 min) (*Malik-Garbi et al., 2019*). The network dynamics arise from distributed actin network assembly and disassembly processes, coupled to myosin-generated forces that drive global network contraction. The network contracts toward a single point, which is located at the center in the symmetric configuration or near the boundary in the polar configuration,

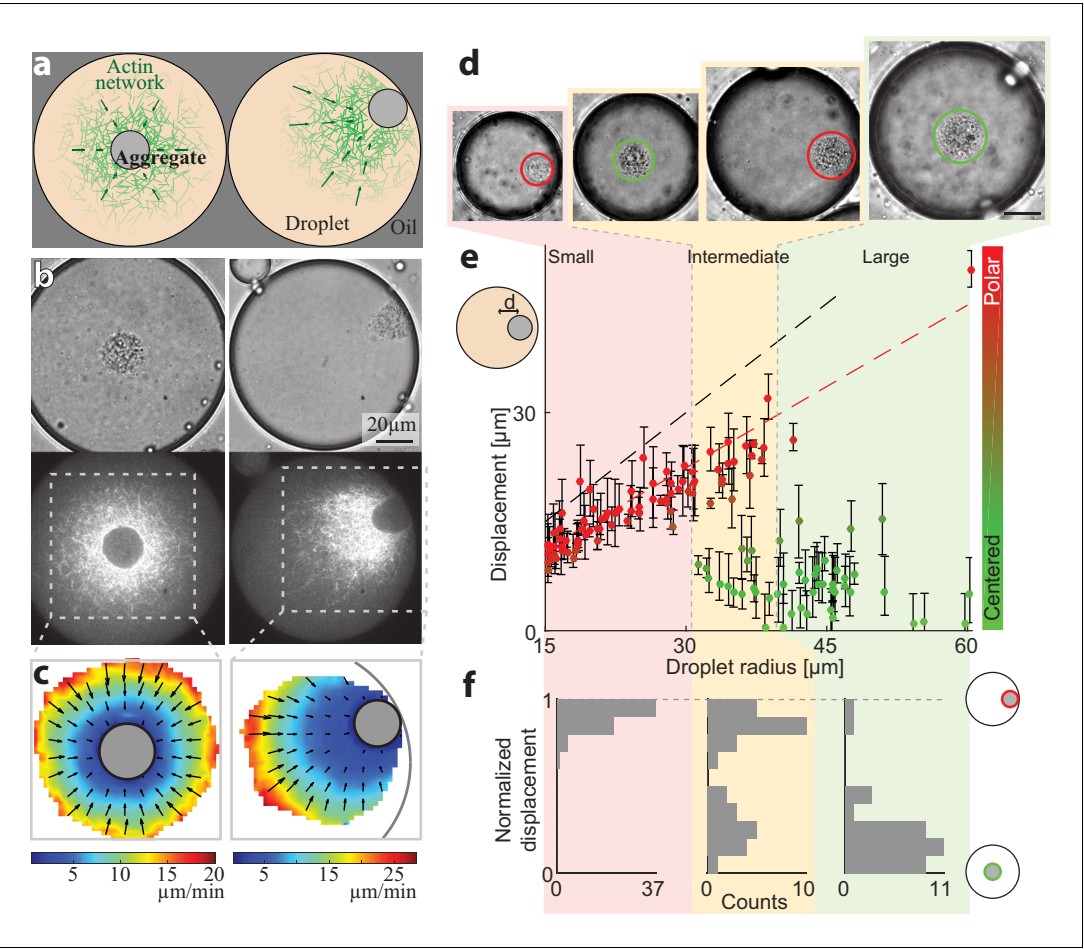

**Figure 1.** Size-dependent localization of the contraction center. (a) Schematic illustration of the two stable configurations of the system: a *symmetric* state with a centered aggregate (left), and a *polar* state in which the aggregate is positioned near the droplet's boundary (right). (b) Bright-field (top) and spinning disk confocal (bottom) images of the equatorial cross section of droplets in a symmetric state (left; *Video 1*) and a polar state (right; *Video 2*). The aggregate is visible both in the bright-field images, and as an exclusion zone surrounded by regions of high actin network density in the florescence images. The actin network is labeled with GFP-Lifeact. (c) The actin network velocity field as determined by correlation analysis of the time lapse movies of the symmetric and polar droplets in (b). The network exhibits contractile flows directed toward the aggregate in both cases. (d–f) The position of the aggregate surrounding the contraction center was determined for a population of droplets of different sizes, 40 min after sample preparation. (d) Bright-field images of droplets of different sizes. The aggregate position in each droplet was determined from the images, and its displacement from the droplet's center was measured (see Materials and methods). (e) The displacement of the aggregate from the center is plotted as a function of droplet radius. The dashed black line marks the droplet radius, and the dashed red line marks the displacement where the aggregate reaches the boundary (droplet radius minus aggregate radius). (f) Histograms of the aggregates localization for droplets in different size ranges: small, intermediate and large (Materials and methods). The distance of the aggregate from the center of the droplet was normalized to be between 0 (centered) and 1 (polar). Small droplet (R < 31 μm) are polar (red; left). Intermediate droplets (31 μm < R < 40 μm) exhibit a bipolar distribution with both symmetric and polar droplets (yellow; center). Large droplet (40 μm < R) are symmetric (green; right).

The online version of this article includes the following figure supplement(s) for figure 1:

**Figure supplement 1.** Contraction center localization and centering dynamics are not dependent on microtubules.

**Figure supplement 2.** Size-dependent localization of the contraction center as a function of time.

generating characteristic centripetal flow patterns around the contraction center (*Figure 1c*). The system is able to reach a dynamic steady-state thanks to the presence of rapid actin network disassembly, which limits the accumulation of network around the contraction center and enables the system to reach a stationary network density distribution (*Malik-Garbi et al., 2019*). We have previously shown that the network contracts at a homogenous, density independent rate (*Malik-Garbi et al., 2019*), with an inward flow velocity that increases linearly in magnitude as a function of distance from the contraction center, and approaches zero on the surface of the aggregate surrounding the contraction center (*Figure 1c*).

The symmetry state of individual droplets was found to be strongly correlated with their size (*Figure 1d–f*). The aggregate is typically centered in larger droplets, whereas in smaller droplets the aggregate is in a polar position near the boundary. The localization of the contraction center in spherical droplets of varying sizes was determined by measuring the displacement of the centroid of the aggregate from the droplet center (Materials and methods). Characterization of the symmetry states 40 min after sample preparation is shown in *Figure 1e*, with small droplets (R < 31 μm) exhibiting a polar state and large droplets (R > 40 μm) predominantly in a symmetric state. Intermediate-sized droplets (31 μm < R < 40 μm) exhibit a bimodal distribution whereby both polar and symmetric droplets are observed. This size-dependent localization pattern depends on actin dynamics, whereas disrupting microtubule assembly with Nocodazole has no effect (*Figure 1—figure supplement 1*). The characteristics of this localization pattern persists for more than an hour (i.e. smaller droplets are polar whereas larger droplets are primarily in a symmetric state), but over time the fraction of droplets in the polar state increases (*Figure 1—figure supplement 2*), suggesting that the centered state is metastable.

## Dynamics of centering and symmetry breaking

To gain more insight into the mechanisms for centering and symmetry breaking of the contraction center, we followed the dynamics of aggregate position over time in different-sized spherical droplets. We measured the 3D positions of the aggregate centroids in time lapse movies (Materials and methods) and analyzed the dynamics of their displacement over time (*Figure 2*). Small droplets were already in a polar state ~5 min after sample preparation. However, we could find droplets with R > 25 μm that were initially symmetric and underwent a symmetry breaking process, whereby the system transitioned from a symmetric state into a polar state (*Figure 2a,b*; *Video 3*). In these cases, the aggregate was initially positioned near the middle of the droplet, exhibiting limited fluctuations, and subsequently, at some moment, started moving in a directional manner toward the droplet boundary (*Figure 2b,f*). The timing of the transition varied between different droplets, whereas the duration of the transition from the center to the boundary was similar, τ = 12 ± 4 min (N = 18). In droplets with R > 35 μm, the symmetric state could remain stable for more than 1 hr (*Figure 2c*).

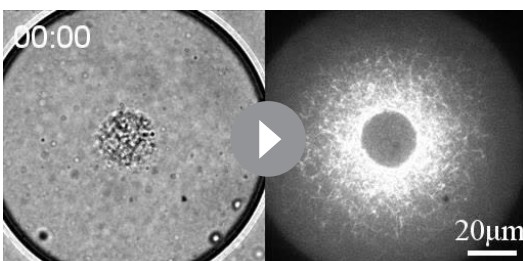

**Video 1.** Steady-state dynamics in a symmetric droplet. This movie shows spinning disc confocal images of a contracting actomyosin network labeled with GFP-Lifeact in a symmetric droplet (*Figure 1b,c*, left). The droplet was squished in a 30μm-high chamber to improve the imaging. Images were taken at the equatorial plane.
https://elifesciences.org/articles/55368#video1

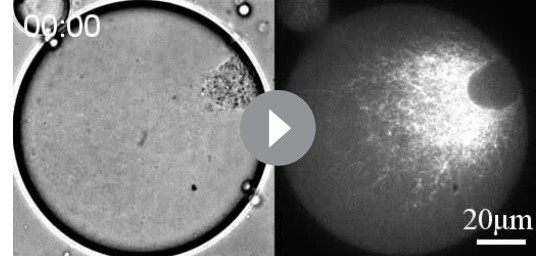

**Video 2.** Steady-state dynamics in a polar droplet. This movie shows spinning disc confocal images of a contracting actomyosin network labeled with GFP-Lifeact in a polar droplet (*Figure 1b,c*, right). The droplet was squished in a 30μm-high chamber to improve the imaging. Images were taken at the equatorial plane.
https://elifesciences.org/articles/55368#video2

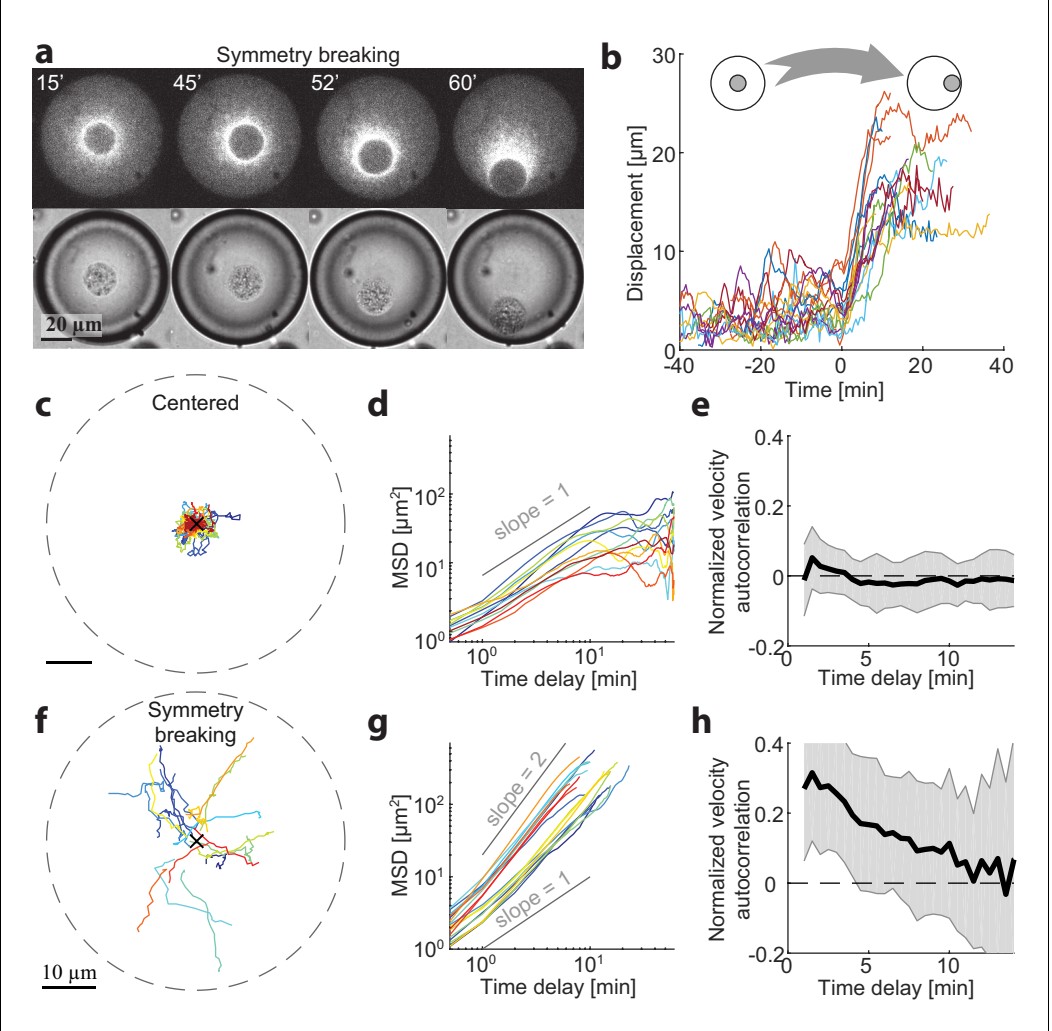

**Figure 2.** Dynamics of centering and symmetry breaking. The dynamics of the contraction center were followed by imaging droplets over time for an hour.(a) Spinning disk confocal (top) and Bright-field (bottom) images from a time lapse movie (*Video 3*) of a droplet that starts in a centered state and breaks symmetry to become polar. (b) The symmetry breaking transition of droplets from a symmetric state to a polar state was characterized in 18 different droplets. The displacement of the aggregate from the center of the droplet is shown as a function of time for the different droplets. Time zero is defined as the onset of symmetry breaking for each droplet (see Materials and methods). (c-h) Analysis of the dynamics of aggregate position as a function of time in droplets in the centered state (c-e; N = 12) and during symmetry breaking (f-h; N = 18). (c,f) Tracks depicting the position of the aggregate in different droplets. (d,g) The mean squared displacement of aggregate positions as a function of time. The droplets in the centered state exhibit confined random fluctuations (d), whereas during symmetry breaking, the movement is directed (g). (e,h) The normalized velocity autocorrelation

$c(\tau) = \frac{\langle \vec{V}(t) \cdot \vec{V}(t+\tau) \rangle}{\langle \vec{V}(t) \cdot \vec{V}(t) \rangle}$ (mean ± STD; averaged over different droplets) is shown for the tracks on the left. The velocity

autocorrelation (for t ≥ 0.5 min) is essentially zero in the centered state (e). During symmetry breaking, the aggregate velocity exhibits a positive correlation for time scales of up to ~ 10 min (h).

The online version of this article includes the following figure supplement(s) for figure 2:

**Figure supplement 1.** Analysis of the dynamics of aggregate position as a function of time in droplets before symmetry breaking.

We characterized the centered state by analyzing the movements of aggregates in droplets that remained symmetric throughout the experiment (1 hr). The aggregates exhibit random movement around the droplet centers. Analysis of this movement shows that the mean squared displacement (MSD) of the aggregate positions in individual droplets scales nearly linearly with time (*Figure 2d*), so that $MSD = \left\langle \Delta \vec{r}(t)^2 \right\rangle \approx 6Dt^{\alpha}$ with $\alpha = 0.93 \pm 0.07$ (Mean ± STD; *N* = 12 droplets) and $D = 0.32 \pm 0.1 \mu m^2 / min$ (Mean ± STD) for t < 5 min. Over longer time scales (t > 10 min), the movement appears confined, as the MSD is bounded and does not continue to increase with time. Thus, the aggregate motion in the centered state can be effectively described as a confined random walk. We can estimate the extent of confinement of the aggregate positions by measuring the mean squared distance from the droplets centers $R_c = \sqrt{\langle d^2 \rangle} = 3.4 \pm 1.6 \mu m$ (Mean ± STD) which is ~10% of the system size. The velocity autocorrelation function decays immediately at the temporal resolution of our measurements (0.5 min; *Figure 2e*), indicating that the aggregates motion is uncorrelated temporally on these time scales.

The characteristics of the symmetry breaking process were analyzed in *N* = 18 droplets that displayed a transition from the symmetric state to a polar state within one hour. Initially, the aggregates exhibited random fluctuations that were similar to the behavior of centered aggregates in the symmetric state (*Figure 2—figure supplement 1*). The onset of symmetry breaking was abrupt, with the aggregate starting to move in a directional manner toward the boundary (*Figure 2b*). This movement was characterized by a mean squared displacement that increased with time as, $MSD = \left\langle \Delta \vec{r}(t)^2 \right\rangle \sim t^{\alpha}$ with $\alpha = 1.5 \pm 0.1$ (Mean ± STD), indicating that the movement of the aggregate at this stage is directed rather than random. The average outward radial velocity during the symmetry breaking process was $V = 1 \pm 1 \mu m / min$ (Mean ± STD). During the symmetry breaking process, the aggregates' velocity becomes temporally correlated as it moved in a directional manner. This is reflected in the velocity autocorrelation function, which in contrast to the symmetric state, exhibits clear temporal correlations over several minutes (*Figure 2h*), comparable to the duration of the symmetry breaking process. Once the aggregate reached the boundary it remained there, in a stable polar state.

## The centered state is stable against large perturbations

To probe the stability of the centered state and gain insight on the centering mechanism, we developed a methodology to apply external forces on the aggregate that allow us to transiently displace the aggregate from the middle of the droplet and then follow its recovery dynamics (*Figure 3*; *Videos 4* and *5*). This is done by introducing micron-sized superparamagnetic beads into the extract mix (*Figure 3—figure supplement 1*; see Materials and methods). During the first few minutes after sample preparation, the magnetic beads are swept together with other particulates in the crude extract into the aggregate that forms around the contraction center. The magnetic beads then allow us to move the contraction center with an external magnet (*Tanimoto et al., 2018*). To that end, we introduce the droplets into rectangular capillaries and use a micromanipulator to position a magnetic needle from the side (*Figure 3a* and *Figure 3—figure supplement 1*). In this configuration, the magnetic force and hence the displacement of the aggregate are nearly horizontal (i.e. parallel to the imaging plane), allowing us to follow the dynamics within a single imaging plane.

To displace an aggregate from the center of a droplet in the symmetric state, we placed a magnetic needle ~200 μm from the droplet (*Figure 3b*). The magnetic force on the beads

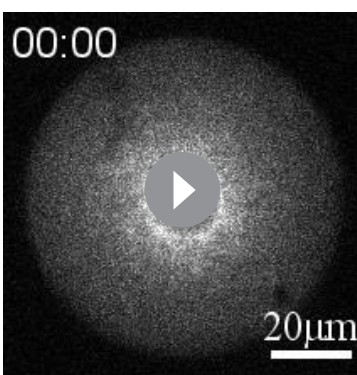

**Video 3.** Symmetry breaking in a droplet that transitions from a symmetric state into a polar state. The dynamics of the actin network labeled with GFP-Lifeact in a spherical droplet were followed by spinning disc confocal time lapse imaging. The movie shows a symmetry breaking process, in which the aggregate moves from the center of the droplet toward the side. https://elifesciences.org/articles/55368#video3

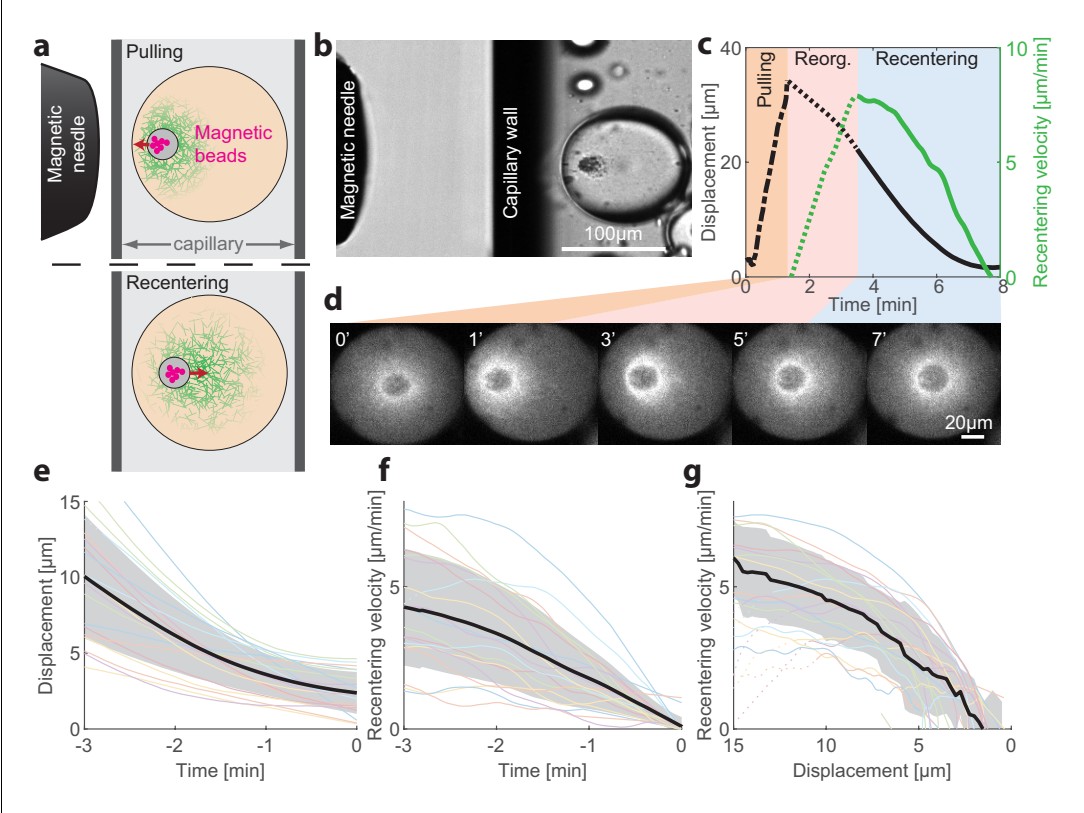

**Figure 3.** Recentering following magnetic perturbations. The stability of the centered state was examined by introducing superparamagnetic beads that become trapped within the aggregate, and applying external magnetic fields to displace the aggregate toward the droplet's boundary. (a) Schematic illustration of the experimental setup. A magnetic needle is introduced from the side, near a droplet placed within a glass capillary with a rectangular cross-section. The magnet exerts a pulling force on the aggregate, displacing it toward the side. The magnetic needle is then removed and the aggregate position is tracked. (b) Bright-field image of a sample, showing the tip of the magnetic needle positioned ~200 µm from a droplet in a capillary. (c) Time traces showing the displacement of the aggregate from the center (black) and its radial velocity (green) during the magnetic perturbation experiment (*Video 4*). During the pulling phase, the magnetic needle is held at a fixed position and the aggregate is displaced toward the side of the droplet ('pulling phase'). After the needle is removed, the aggregate starts moving and reaches a maximal inward velocity after ~2 min ('reorganization phase'). The recentering continues with the aggregate gradually slowing down and eventually stopping as it approaches the droplet's center ('recentering phase'). (d) Spinning-disk confocal images of the actin network labeled with GFP-Lifeact from a timelapse movie showing the recentering of the contraction center following a magnetic perturbation (*Video 4*). (e) The position of the aggregate as a function of time after removing the magnet is shown for different droplets. (f) The recentering velocity as a function of time is shown for different droplets. (g) The recentering velocity is plotted as a function of the displacement from the droplet center. (e-g) Dotted lines - reorganization phase, solid lines - recentering phase. The thick black line depicts the average over the collection of droplets (shaded region - standard deviation).

The online version of this article includes the following figure supplement(s) for figure 3:

**Figure supplement 1.** Magnetic pulling experimental setup.

**Figure supplement 2.** Transition to a polar state following a magnetic perturbation.

**Figure supplement 3.** Asymmetric actin network distribution during recentering.

pulls the aggregate toward the needle. As the aggregate is connected to the cytoskeletal network surrounding it, this causes the complex of the aggregate and the surrounding network to move from the middle of the droplet toward the side. The magnetic needle was kept in place for ~1 min while monitoring the aggregate's position. We estimate that the net pulling force acting on the aggregate is of the order of ~50 pN (Materials and methods). The needle is then removed, and the recovery dynamics of the aggregate are followed. Typically, we observed that shortly after removing the magnetic needle, the aggregate recentered, moving in a directional manner back toward the center of the droplet (26 out of 31 droplets examined; *Figure 3*, *Video 4*). Occasionally, we observed droplets that transitioned into a polar state after the perturbation (5 out of 31 droplets examined; *Figure 3— figure supplement 2*, *Video 5*).

We followed the recentering dynamics of the aggregates following large displacements in $N = 19$ droplets (*Figure 3*), measuring the recentering velocity as a function of time (*Figure 3c,f*) and as a function of distance from the center of the droplet (*Figure 3g*). We find that the aggregates move toward the center of the droplet and resettle there in a symmetric configuration. The centering velocity reaches a peak value of 5–10 µm/min after an initial reorganization phase. Subsequently, the recentering process proceeds at a velocity that decreases as a function of the displacement of the aggregate from the droplet center in a characteristic concave-down fashion, which is similar among different droplets (*Figure 3g*). These results show that the centered position of the aggregate is actively maintained by a centering force and is stable against large perturbations.

The recentering process is accompanied by a dynamic reorganization of the actin network (*Figure 3d* and *Figure 3—figure supplement 3*). While the contractile network flows persist throughout the recentering process, the network density distribution and flow pattern undergo substantial rearrangements. During the reorganization phase, following the large displacement of the aggregate from the droplet center, the network distribution around the aggregate becomes skewed with a higher density toward the middle of the droplet. As the aggregate moves toward the center of the droplet, the asymmetry in the network distribution becomes less prominent until eventually, when the aggregate reaches the center of the droplet, the network distribution becomes symmetric again (*Figure 3—figure supplement 3*).

## Hydrodynamic mechanism for centering

To understand the centering mechanism, we model our system as a two-phase system made up of an active actomyosin network immersed in the surrounding fluid (cytosol), both enclosed within a spherical droplet with the aggregate as an excluded region (*Figure 4a*). Mathematically, we describe the system using dynamical equations for the actin network density ($\rho$) and the coupled flows of the network ($V$) and the fluid cytosol ($U$; see Appendix 1). Many studies (reviewed in *Mogilner and Manhart, 2018*; see also *Shamipour et al., 2019*) demonstrated that the cytosol can be considered as a viscous fluid that flows in the cell with a low Reynolds number, and squeezes through effective pores formed by the cytoskeletal mesh. Physically, the relative movements of the fluid cytosol and the cytoskeletal mesh lead to the so-called Darcy friction, which is proportional to the relative local movement between the mesh and the cytosol. Given the network density and movement rates, a well-defined system of equations (*Mogilner and Manhart, 2018*) allows one to calculate the Darcy friction and the resulting pressure distribution and flow in the cytosol. The Darcy friction between the contracting network and the fluid cytosol depends on the relative velocity between them, the fluid viscosity, and the network permeability coefficient which is a function of the network density (*Charras et al., 2005*; *Schmidt et al., 1989*). Based on our previous work (*Malik-Garbi et al., 2019*), we assume that the actin network exhibits centripetal flow with a speed that increases linearly with the distance from the aggregate (*Figure 1c*), and that the network turns over with constant assembly and disassembly rates. Two fundamental laws – conservation of mass and momentum – govern the dynamics and mechanics of the network and fluid flow. We numerically solve the respective coupled equations and obtain the actin network density distribution, the fluid velocity and pressure distributions, and the position of the aggregate (Appendix 1). The net centering force on the aggregate is obtained by integrating the friction forces due to the relative movement between the contracting network and the fluid phase (Appendix 1).

To estimate the effective force on the aggregate, we performed simulations in which the aggregate is held fixed in place at a given position, and the network is allowed to reach a dynamic steady-state for this configuration (*Figure 4b,c*). A 'fountain'-like cytosolic flow emerges when the aggregate is decentered: if the aggregate is shifted to the left (*Figure 4b,c*), more network accumulates to its right and this body of network flows to the left. The cytosol is pulled to the left with the contracting network, and then, due to the incompressibility, escapes around the aggregate to the left curving away from the aggregate. The cytosol returns to the right near the boundaries of the droplets, where the network density is low and the resistance to fluid flow is smaller (*Figure 4c*).

We calculate the net hydrodynamic force on the aggregate by integrating the Darcy friction force over the whole droplet, as a function of aggregate position and droplet radius (*Figure 4d*). We find that the hydrodynamic centering force behaves like a Hookean spring with a force that increases linearly as a function of the displacement from the droplet center, and an effective spring constant that scales with the volume of the droplet (*Figure 4d,e*). This scaling arises because the network density

depends weakly on the radius, so each volume element contributes a certain force and the net force scales with the droplet's volume. The hydrodynamic centering force is strongly dependent on the contraction rate of the actin network: faster contraction increases the relative movement between the network and the surrounding fluid, and hence enhances the Darcy friction forces which generate the centering force (*Figure 4f*).

Intuitively, the appearance of a centering force can be understood as follows. The contracting network flows through the cytosol. When the aggregate is displaced from the center, for example to the left, then the network distribution becomes skewed with more network on the right (it has more space to assemble on the right) (*Figure 4b*). As the network permeability decreases with network density, the Darcy friction force is higher on the right, and also the force is integrated over a greater volume on the right. The direction of the force from every element of volume is opposed to the network flow (*Figure 4c*), so the net force on the aggregate will be directed toward the droplet center. Importantly, this centering force does not involve a direct interaction with the droplet boundary (e.g. push/pull). Rather, the centering force arises from hydrodynamic interactions of the network with the fluid cytosol at low Reynolds number. The presence of this effective centering force also explains the confined nature of the aggregate motion observed in the centered state (*Figure 2c,d*).

To model the dynamics of the magnetic recentering experiments (*Figure 3*), we performed additional simulations in which the aggregate was free to move (*Figure 4g–h,k*). To emulate the experimental configuration, we assume initial conditions in which the network distribution is equal to the steady-state distribution in the centered state that is then displaced toward the side (*Figure 4g*; Appendix 1). We simulate the evolution of the system by numerically solving the coupled equations for the network and fluid dynamics iteratively, whereby at each step the aggregate moves in response to the net force acting on it (details in Appendix 1). The simulated aggregate recenters to the middle of the droplet with a centripetal velocity that decreases as a function of its displacement from the droplet center. The movement of the aggregate in the simulation is correlated with the extent of asymmetry in the actin network distribution, which agrees with our experimental observations (*Figure 3—figure supplement 3*).

The model further predicts that the centering dynamics will be strongly enhanced by the rate of network contraction, but will be nearly independent of the fluid viscosity and the meshwork assembly and disassembly rates (*Figure 4—figure supplement 1*). These trends can be easily understood intuitively; both the hydrodynamic driving force for recentering and the opposing drag are dependent on the fluid viscosity and the meshwork permeability in a similar manner, so the effect of changing these properties cancels out and as a consequence has little influence on the centering

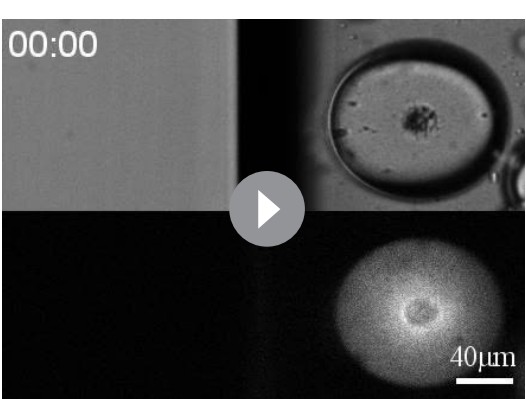

**Video 4.** Recentering dynamics following a magnetic perturbation. This movie shows spinning disc confocal and bright-field images of a magnetic perturbation experiment (*Figure 4b–d*). The aggregate was pulled from the center using a magnetic needle, and subsequently recentered after the magnet is removed.
https://elifesciences.org/articles/55368#video4

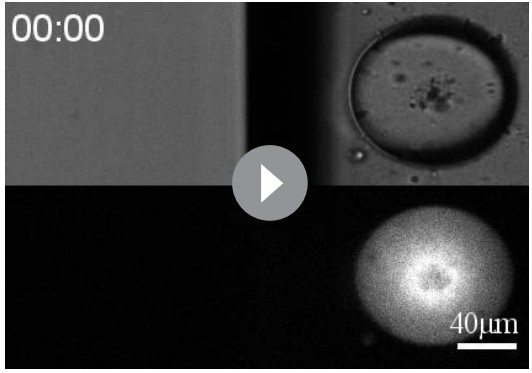

**Video 5.** Transition to a polar state following a magnetic perturbation. This movie shows spinning disc confocal and bright-field images of a magnetic perturbation experiment in which the droplet became polar following the perturbation (*Figure 3—figure supplement 2*). The aggregate was pulled from the center using a magnetic needle, and subsequently continued moving in the same direction until it reached the boundary.
https://elifesciences.org/articles/55368#video5

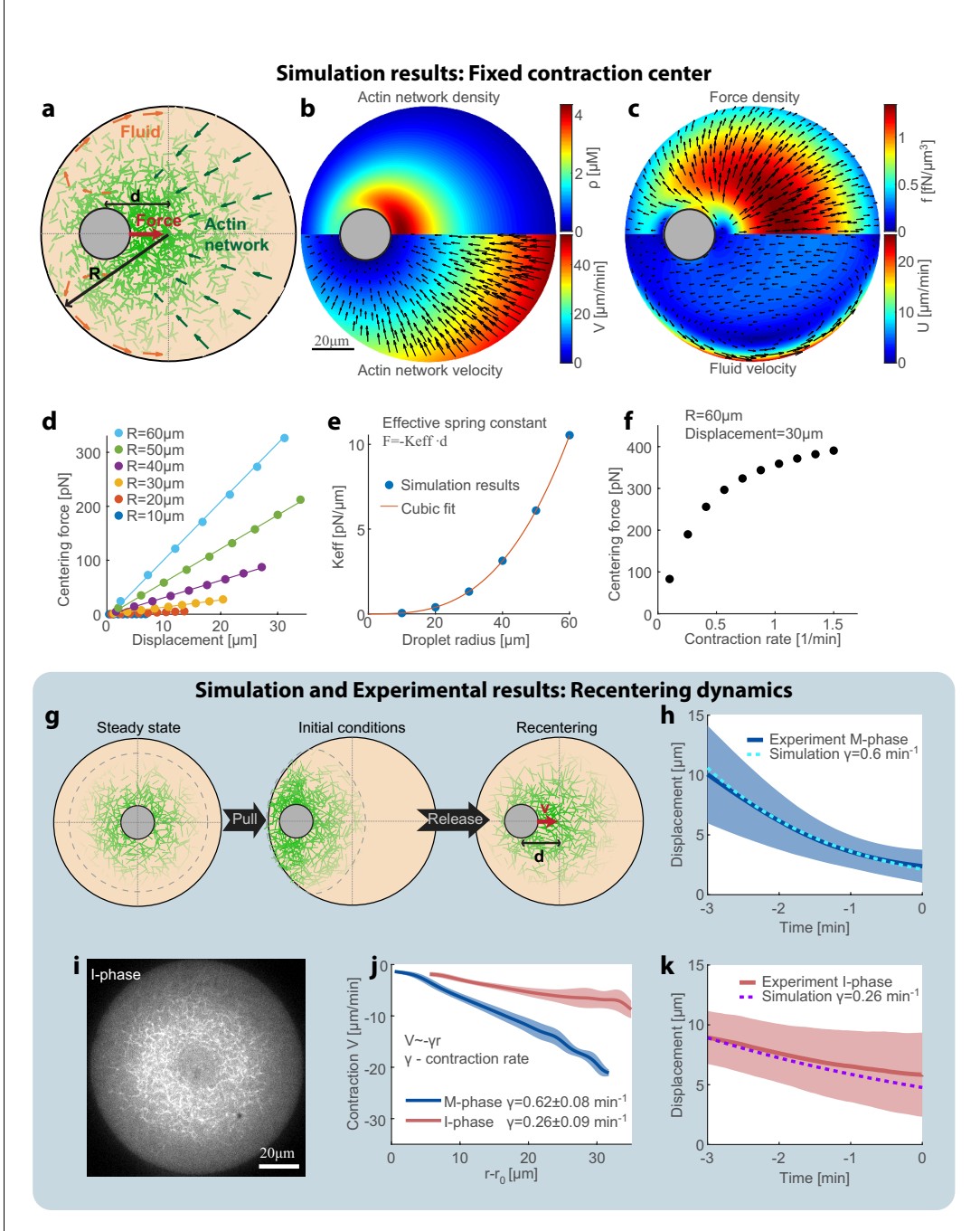

**Figure 4.** Recentering model: friction between the contracting network and the fluid cytosol generates a centering force. (**a**) Schematic illustration of the model. The system is modeled as a two-phase system composed of a contracting actin network and the surrounding fluid. (**b–f**) Simulation results of the model when the aggregate position is held fixed (Appendix 1). The simulation results for the steady-state actin network density and flow (**b**) and the surrounding fluid flow and force density (**c**) are shown. (**d**) The centering force is calculated from the simulation as the net force exerted on the aggregate as a function of the displacement from the droplet center at steady-state. The net centering force is plotted as a function of the displacement for droplets of different sizes (dots- simulations results; line- linear fit). (**e**) The effective spring constant of the centering force, determined from the slope of the linear fit to the simulation results in (**d**), is plotted as a function of the droplet radius (R). The effective spring constant increases as $R^3$. (**f**) The centering force is plotted as a function of the network contraction rate. The centering for was calculated for a displacement of 30 μm in a droplet with a radius of R = 60 μm. The centering force increases with network contraction. (**g–k**) In the dynamic model (Appendix 1), the aggregate position is part of the dynamic variables of the system and moves according to the net force acting on it. (**g**) Schematic illustration of the dynamic model used to model the magnetic perturbation experiments. The initial conditions are obtained by displacing a centered steady-state configuration toward the side. Subsequently, the dynamics of the system lead to recentering of the aggregate. (**h**) The simulation results for the displacement of the

*Figure 4 continued on next page*

*Figure 4 continued*

aggregate as a function of time (dashed line) are compared to the average displacement determined experimentally in M-phase extracts (standard condition) (blue line; mean ± STD; *Figure 3e*). (i) Spinning-disk confocal image of the equatorial cross section of an I-phase extract droplet. The actin network is labeled with GFP-Lifeact. (j) Network contraction velocity measured as a function of distance from the inner network boundary, for M-phase and I-phase extracts (mean ± STD). The contraction rate, γ, is determined from the slopes of linear fits to the data (*Malik-Garbi et al., 2019*). (k) The simulation prediction for the displacement of the aggregate as a function of time (dashed line) are compared to the average displacement determined experimentally in I-phase extracts (red line; mean ± STD). The observed average I-phase contraction rate is used as an input for the simulation which contains no additional fit parameters.

The online version of this article includes the following figure supplement(s) for figure 4:

**Figure supplement 1.** The network contraction rate is the main control parameter for the hydrodynamic centering mechanism.
**Figure supplement 2.** Aggregate recentering velocity in M-phase and I-phase extracts.
**Figure supplement 3.** Recentering dynamics in ActA-supplemented extract.
**Figure supplement 4.** Cell-cycle dependence of the localization of the contraction center.

dynamics. In contrast, the contraction rate of the network is directly related to the magnitude of the hydrodynamic centering force, but does not affect the resisting drag force. As such, a decrease (increase) in the network contraction rate is expected to cause a corresponding slow down (speed up) of the centering process (*Figure 4—figure supplement 1*).

To test these predictions experimentally, we modulated the behavior of the system by changing the cell cycle state of the extract or by varying the system's composition, and examined the relation between the centering dynamics and the network contraction rate (*Figure 4i-k*, *Figure 4—figure supplement 2* and *Figure 4—figure supplement 3*). Cycling the meiotic (M-phase) extract into interphase (I-phase) has a dramatic influence on actin-myosin dynamics (*Field et al., 2011*) and results in a sparser network with a ~ 3 fold slower contraction rate (*Figure 4i,j*). We find that this substantial reduction in the network contraction rate is accompanied by slower centering dynamics. These observations are in agreement with the predictions of the model, taking into account the measured changes in the network contraction rate with no additional fit parameters (*Figure 4j,k* and *Figure 4—figure supplement 2*). Similar results are obtained with networks supplemented with ActA that promotes branched filament nucleation, and has recently been shown to induce changes in network behavior, and in particular slow down network contraction (*Malik-Garbi et al., 2019*). Again, we find that the slower contraction rate is accompanied by slower centering dynamics as predicted by the model (*Figure 4—figure supplement 3*). While both perturbations likely influence the meshwork permeability, the effect on the centering dynamics can be quantitatively accounted for solely based on the observed reduction in the contraction rate. Moreover, the weaker centering force in interphase-extracts also leads to cell-cycle dependent changes in the localization pattern of the contraction center, with more droplets assuming a polar configuration (*Figure 4—figure supplement 4*).

## Symmetry-breaking is induced by interaction with the boundary

The hydrodynamic interaction generates a centering force that explains the stability of the centered state. But how does the system break symmetry and become polar? Why is the symmetry state of the system size-dependent? We posit that the decentering is driven by an attractive interaction between the actin network and the droplet boundary, for example by crosslinking of the network filaments to proteins on the boundary. In this scenario, the symmetry state of the system is determined by a competition between the hydrodynamic centering force and the attractive interaction with the boundary (*Figure 5a*).

To test this idea experimentally, we modulated the interaction between the actin network and the boundary by adding low concentrations of Bodipy-conjugated ActA, which is an actin nucleation promoting factor that has been engineered to localize to the water-oil interface (*Abu Shah and Keren, 2014*; *Tan et al., 2018*). The presence of ActA at the interface activates the actin nucleator Arp2/3 and promotes the nucleation of actin filaments at the surface. We reasoned that the addition of low levels of ActA (much lower than the amounts required to form a continuous cortical network [*Abu Shah and Keren, 2014*]) would have a negligible influence on the bulk actin network, but would increase the likeliness of transient network attachment to the boundary via the surface nucleated filaments. We find that indeed as the concentration of ActA-bodipy is increased, the system

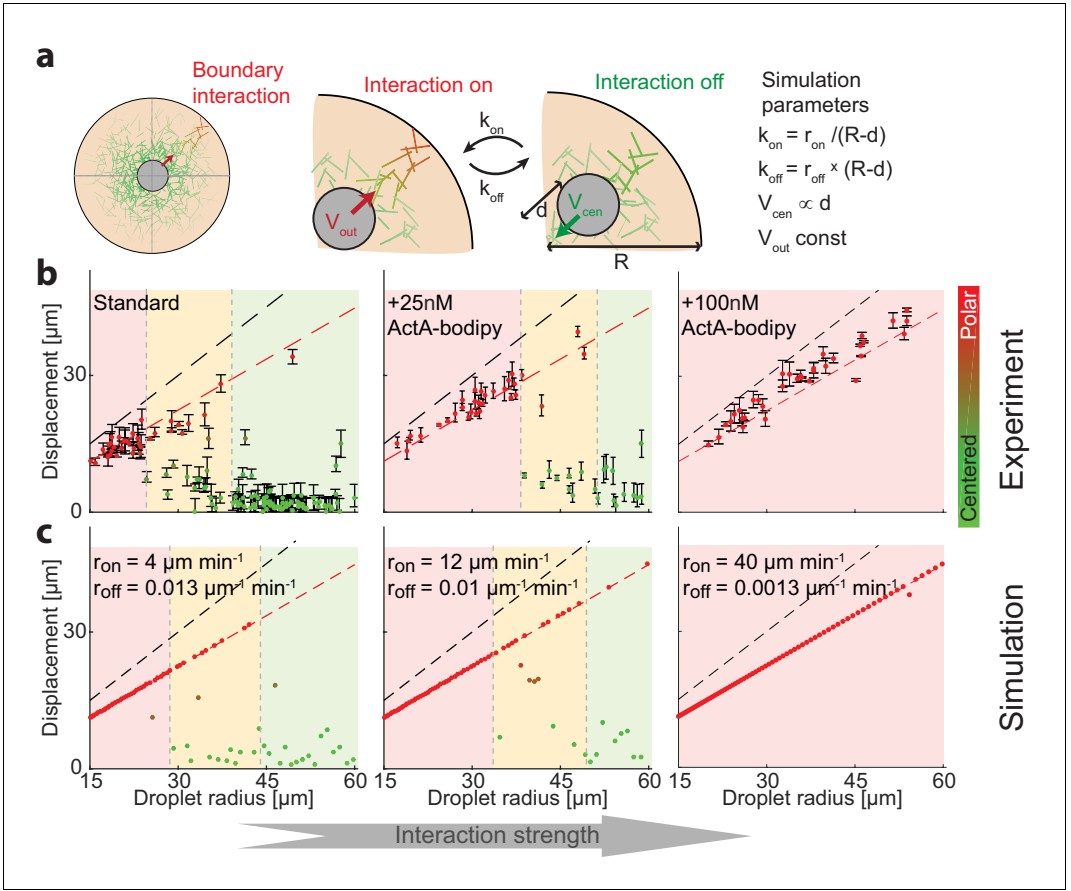

**Figure 5.** Symmetry breaking is induced by an attractive interaction with the boundary. (**a**) Schematic illustration of the system which contains a bulk contracting network that has some attractive interaction with the boundary. Experimentally, the interaction with the boundary can be enhanced by adding increasing concentrations of Bodipy-conjugated ActA that localizes to the water-oil interface and nucleates actin filaments there (**Abu Shah and Keren, 2014**). In simulations, we introduce a clutch that can engage and disengage stochastically, connecting the contracting actin network with the boundary (Appendix 2). The competition between the recentering force that drives the aggregate to the center and the attractive interaction with the boundary, leads to size-dependent localization. (**b**) The aggregate displacement as a function of droplet radius is shown for standard conditions (left), and samples supplemented with 25 nM (center) or 100 nM (right) Bodipy-ActA. The colored regions depict the different size ranges: small droplets that are primarily polar (red), intermediate range with both polar and symmetric droplets (yellow), and large droplets which are mostly centered (green). The dashed black line marks the droplet radius, and the dashed red line marks the displacement where the aggregate reaches the boundary (droplet radius minus aggregate radius). As the interaction with the boundary is enhanced by increasing ActA concentrations, larger droplets are found in a polar state. (**c**) Simulation results for the aggregate displacement as a function of droplet radius. The interaction with the boundary is enhanced by increasing the clutch engagement rate ($k_{on}$) and decreasing the disengagement rate ($k_{off}$), resulting in changes in the size-dependent localization of the aggregates that resemble the changes observed experimentally in (**b**).

The online version of this article includes the following figure supplement(s) for figure 5:

**Figure supplement 1.** Size-dependent localization of the contraction center is not influenced by low levels of bulk ActA.

**Figure supplement 2.** Sample geometry induces anisotropy in the localization of the aggregate in the polar state.

shifts toward a polar configuration (**Figure 5b**). These changes were related to the localization of ActA at the interface, as control experiments with the same concentration of ActA in the cytoplasm did not induce symmetry breaking (**Figure 5—figure supplement 1**).

To model the size-dependent symmetry state of the system, we considered a simple effective model which incorporates the hydrodynamic centering force (**Figure 4d,e**) and also assumes that the

network can stochastically engage with the boundary (Appendix 2). While the exact nature of the interaction between the bulk actin network and the boundary is not known, we assume the network interacts by a stochastic 'clutch' mechanism, with a characteristic on-rate ($k_{on}$) and off-rate ($k_{off}$), that when engaged pulls the aggregate to the closest boundary. We further assume that the probability to engage with the boundary depends on the actin network density near the boundary, reasoning that the on and off-rates should increase or decrease, respectively, as a function of the network density close to the boundary. Since the network density falls off as a function of the distance from the contraction center, we assume the on-rate increases when the aggregate is closer to the boundary (since the network density near the boundary becomes larger) while the off rate decreases. Specifically, for simplicity, we assume that the off-rate is proportional to the distance between the aggregate and the closest boundary, while the on-rate is inversely proportional to that distance. Finally, we assume that when the aggregate is engaged, it moves at a constant speed toward the closest boundary, to mimic the directed motion we observe during symmetry breaking (*Figure 2*).

This simple model recapitulates the size-dependent symmetry states in the system, predicting that small droplets will be polar and large droplets will be centered, with a transition zone for intermediate-sized droplets. The centered state in this model is metastable; while larger droplets tend to stay centered for long periods because the probability to engage the clutch is lower, eventually, due to the stochastic nature of the surface interaction, they can break symmetry. As a result, the size-dependence of the symmetry states in the system evolves over time, with larger and larger droplets becoming polar as observed experimentally (*Figure 1—figure supplement 2*). The size-dependent localization pattern that arises from a competition between the hydrodynamic centering and the surface interactions can be tuned by varying the properties of the contracting network or the boundary. For example, when we enhance the interaction with the boundary by increasing the probability of clutch engagement and/or decreasing the probability for disengagement, to mimic the increase in cortical actin following the addition of ActA at the interface (*Figure 5b*), the model predicts that the transition between the symmetry states occurs at larger droplet sizes, which is aligned with our experimental observations (*Figure 5c*). We can further test the model predictions by varying the geometry of the droplets. The model predicts that for non-spherical droplets, symmetry breaking will be biased toward the boundary closest to the droplet's center. Indeed, we find that in squished, pancake-shaped droplets, symmetry breaking occurs preferentially toward the top or bottom interface, in contrast with the more homogenous angular distribution in spherical droplets (*Figure 5—figure supplement 2*).

Note, that the alternative to the clutch model would be assuming the presence of a constant interaction force with the boundary that is a function of the distance between the aggregate and the boundary. However, such a model would predict that following the magnetic force pulling experiments, which bring the aggregate close to the boundary, the interaction force with the boundary would either be stronger than the hydrodynamic centering force, in which case we would not expect the aggregate to recenter, or weaker than the hydrodynamic centering force, whereby decentering would never take place. This contradicts our observations (*Figures 2* and *3*). The clutch model, on the other hand, is consistent with the system's behavior following the magnetic perturbations, and also accounts for the observed time-dependence: 1) Larger droplets remain centered over a finite time interval because the frequency of the clutch engagement decreases with radius. 2) After sufficient time, decentering occurs even in large droplets, because when the clutch is engaged, the interaction with the boundary overcomes the hydrodynamic force. While the quantitative predictions of our symmetry breaking model (Appendix 2) depend on the specific realization of the surface interaction and the parameters used, the qualitative localization pattern does not. The essential feature of the surface interaction is their transient nature, whereas any model that assumes a continuous interaction is unable to account for the observed phenomenology (i.e. a bimodal distribution of cells in a centered and polar configuration, where the centered state is robust against large perturbations yet meta-stable over time).

## Discussion

We developed a model system to study actin-based localization in artificial cells. The system exhibits two stable configurations; a centered configuration in which the contraction center is actively maintained at the middle of the cell and a decentered one where the contraction center is near the

boundary. While the contracting network dynamics are largely determined by the interplay between internal forces, namely the myosin induced contractile force and the opposing force which is primarily due to the viscosity of the network (*Malik-Garbi et al., 2019*), the size-dependent localization of the contraction center is determined by the smaller friction forces between the network and the fluid it is immersed in and the interactions between the network and the droplet boundary. The observed dependence of intracellular localization on cell size and cell cycle state can have important implications for early embryonic development, where cycles of rapid cell division with limited growth typically lead to a rapid decrease in cell size. Our results illustrate that changes in cell size and cell-cycle state can have a substantial effect on the symmetry state of the system. In particular, our results suggest that simply reaching a certain cell size can be sufficient for inducing a transition from a symmetric configuration to symmetry breaking at a certain developmental stage.

By combining experimental measurements and theoretical modeling, we show that a simple and robust hydrodynamic centering mechanism is responsible for actively maintaining the contraction center at the middle of the droplet (*Figure 4*). The hydrodynamic centering is based on the presence of a persistent centripetal network flow, that is subject to Darcy friction with the surrounding cytoplasm. A net centering force arises from an imbalance in the friction forces when the network distribution is asymmetric, providing an indirect way to sense the shape of the cell as the boundary of the domain in which the reaction-diffusion-convection dynamics of the network take place. The net force that arises can lead to centering of cellular components even without generating appreciable intracellular cytoplasmic flows (*Figure 4c*). Since the hydrodynamic centering force scales with cell volume (*Figure 4e*), the contribution of the hydrodynamic centering mechanism is expected to become more prominent in large cells such as oocytes.

The main control parameter for the centering mechanism is the network contraction rate (*Figure 4—figure supplement 1*), a property that can be modulated by changing the properties and concentrations of various cytoskeletal regulators (*Malik-Garbi et al., 2019*), e.g. as a function of the cell cycle or developmental state of the cell. Importantly, the centering mechanism is extremely robust; the network naturally self-organizes into an asymmetric configuration when the contraction center is skewed, allowing cells to dynamically respond and adapt to changes in their shape, and the centering dynamics are insensitive to variations in the structure of the meshwork and/or the viscosity of the cytoplasm (*Figure 4—figure supplement 1*). Furthermore, unlike other cytoskeletal-based cellular centering mechanisms (*Tanimoto et al., 2016*; *Wu et al., 2017*), the hydrodynamic centering mechanism does not involve any direct interactions with the boundary. Notably, the hydrodynamic centering mechanism proposed is generic; the centering force does not depend on the specific molecular components involved. Rather, the same centering mechanism would function in the presence of continuous flow of any cellular component. The important features are the presence of persistent flow patterns that generate directed Darcy friction forces, and a dynamic density of the flowing components that changes in response to its cellular localization.

Our system can also break symmetry, and transition from the centered state to an asymmetric polar state in a size-dependent manner. Our results suggest that this transition reflects a competition between the hydrodynamic centering force and attractive interactions with the boundary, that can be enhanced by the presence of a cortical actin layer at the interface (*Figure 5*). While the hydrodynamic drag originating from the bulk cytoplasmic flows generates a continuous net centering force, the attractive interaction with the boundary (likely generated when a part of the network becomes anchored to the interface) can be transient – as it depends on the presence of a mechanical link between the contracting network and the boundary. Filament turnover and active forces that cause local network rupture events can break this link and thwart the pulling forces towards the boundary. The detachment of the anchor between the contracting network and the interface reverses the direction of force exerted on the aggregate: an anchored contracting network pulls the aggregate toward the boundary, whereas an unanchored network pushes the contraction center away from the closest boundary via the hydrodynamic centering mechanism. Our analysis (Appendix 2) indicates that the transient character of the surface interaction is essential for the time and size-dependent localization pattern observed, facilitating a centered state that is robust against large fluctuations, yet susceptible to symmetry breaking over time.

While the exact nature of the surface interactions is not well characterized, our results suggest that the tendency to polarize in smaller droplets arises primarily from the higher probability for the contracting network to engage with the boundary in these droplets. We model the interaction

between the contracting network and the boundary as a transient 'clutch'. While the actual interactions are likely more complicated than assumed in this simplified model, the agreement between our experimental results under different conditions and the predictions of this model, suggest that the simplified model captures, at least qualitatively, the main features responsible for the size-dependent centering and decentering of the contraction center. Importantly, the strong dependence of the centering efficiency on the contraction rate implies that cells can regulate the localization of cellular components via modulations of the actin network dynamics. In particular, our results show that changes in the cell cycle state which modulate the contraction rate have a large influence on the observed localization pattern (*Figure 4—figure supplement 4*).

This novel symmetry breaking mechanism joins an array of different actin-myosin-based symmetry breaking mechanisms that have been previously discovered (*van Oudenaarden and Theriot, 1999*; *van der Gucht et al., 2005*; *Kozlov and Mogilner, 2007*; *Mullins, 2010*; *Abu Shah and Keren, 2014*; *Barnhart et al., 2015*), demonstrating yet another way in which these active networks can break symmetry and generate large scale motion. In all these examples, the symmetry breaking mechanisms reflect an inherent mechanical instability in the system, which can be biased by the presence of directional cues, allowing cells to polarize in response to various signals (*Mullins, 2010*). Similarly, in our system the transition from a centered state to a polar one is spontaneous and occurs in a random direction, yet the same mechanism could allow cells to respond to a directional signal, for example if nucleation of cortical actin was enhanced locally, in a particular region of the cell boundary, the system would polarize in that direction.

The development of a simplified reconstituted model system, allowed us to discover and characterize in detail novel actin-based mechanisms for cellular localization. The characteristics of our in vitro system, such as the size of the cells and the typical velocities, are comparable to those found in oocytes and large embryonic cells, so we expect that the localization mechanisms explored here can be relevant for localization of various sub-cellular components in vivo. To assess the contribution of the actin-based hydrodynamic mechanism proposed here for the localization of organelles within living cells will require measuring the intracellular forces on these components in vivo (*Tanimoto et al., 2018*), and examining how local or global disruptions of the actin network flow influence the forces and the resulting localization patterns. Since the mechanisms for cellular localization in vivo are diverse, especially during early embryogenesis, and also involve additional force-generating systems such as the microtubule cytoskeleton (*Gundersen and Worman, 2013*; *Mitchison, 2012*; *Xie and Minc, 2020*; *Wühr et al., 2009*; *Mogessie et al., 2018*), we expect the contribution of the hydrodynamic centering to vary considerably among organisms and between different cellular contexts. For example, nuclear positioning in *Xenopus* embryos appears to be primarily microtubule-dependent (*Wühr et al., 2009*), and hence is unlikely to involve the actin-based mechanisms discussed here. In contrast, localization of cellular components in other systems such as mammalian oocytes is known to be actin-dependent (*Xie and Minc, 2020*; *Almonacid et al., 2018*; *Uraji et al., 2018*). In these systems, we expect the mechanisms discussed here will contribute to the positioning of organelles when the actin networks exhibit large-scale flows. While bulk actin network flows are often masked by cortical dynamics and hence difficult to detect (*Field and Lénárt, 2011*), they have been observed in various systems including starfish oocytes (*Lénárt et al., 2005*) and zebrafish oocytes (*Shamipour et al., 2019*; *Ierushalmi and Keren, 2019*).

The involvement of persistently flowing actin networks in cellular centering and decentering is seen in diverse cellular contexts. Examples include the localization of the nucleus in migrating cells that transition from a centered state in stationary cells to a polar localization in motile cells in fibroblasts (*Gomes et al., 2005*) and keratocytes (*Yam et al., 2007*). Note also that this decentering and polarization is often the hallmark of motility initiation in cells, both in 2D (*Barnhart et al., 2015*) and in 3D (*Callan-Jones and Voituriez, 2016*), adding to the significance of the mechanism that we uncovered. Future research will show if Darcy forces could be part of the motility initiation phenomena in cells.

An important feature of the localization mechanisms based on actin network flow is their ability to operate across scales and drive transport even over macroscopic scales. The transport of cellular components can be directly driven by the network flow, as seen for example during chromosome congression in star fish oocytes (*Lénárt et al., 2005*), or indirectly via friction based mechanisms as observed recently during ooplasm segregation in zebrafish oocytes (*Shamipour et al., 2019*; *Ierushalmi and Keren, 2019*). In our system, the movement of the aggregate during decentering is

directly coupled to the flow of the contracting actin network, whereas the centering is indirectly mediated by a hydrodynamic interaction between the contracting network and the surrounding cytoplasm.

There is an analogy between the localization mechanism we observed and a general class of mechanisms that can regulate switches between centering and decentering, based on shifting the balance between the interaction of the network periphery with the cell boundary and forces that act along the network length (*Tanimoto et al., 2016*; *Zhu et al., 2010*; *Fogelson and Mogilner, 2018*). The novelty of our mechanism is that the centering force along the network length is hydrodynamic in nature. There was, in fact, a previous proposal that cytoplasmic flow generated by drag from dynein-driven cargo on astral microtubules can position organelles in large cells (*Shinar et al., 2011*; *Niwayama et al., 2011*). Involvement of an actin-network-generated cytoplasmic flow in decentering of meiotic spindle in mouse oocytes was also proposed (*Yi et al., 2013*). In general, the appreciation for the presence of cytoplasmic flow in cells, driven by either actin-myosin (*Keren et al., 2009*) or microtubule-kinesin-dynein networks (*Monteith et al., 2016*), is increasing. The contribution of our study is that we demonstrate the significance of such flow for subcellular localization in a minimal in vitro system.

The mechanistic understanding of the processes responsible for the localization of cellular components in vivo and the force generation involved, especially in large cells, is still in many cases surprisingly limited (*Xie and Minc, 2020*). The development of in vitro model systems to study centering and decentering mechanisms in cells, as exemplified by this work, provides important insights to understand the complex dynamics that determine the internal cellular organization, which is an essential step toward deciphering the underlying operation principles of the living cell.

## Materials and methods

### Cell extracts, Proteins and Reagents

Concentrated M-phase extracts were prepared from freshly laid *Xenopus laevis* eggs as previously described (*Abu Shah and Keren, 2014*; *Malik-Garbi et al., 2019*; *Abu-Shah et al., 2014*). Briefly, *Xenopus* frogs were injected with hormones to induce ovulation and laying of unfertilized eggs for extract preparation. The eggs from the different frogs were collected and washed with 1X MMR (100 mM NaCl, 2 mM KCl, 1 mM $MgCl_2$, 2 mM $CaCl_2$, 0.1 mM EDTA, 5 mM Hepes, pH 7.8, 16°C). The jelly envelope surrounding the eggs was dissolved using 2% cysteine solution (in 100 mM KCl, 2 mM $MgCl_2$, and 0.1 mM $CaCl_2$, pH 7.8–7.9). Finally, eggs were washed with CSF-XB (10 mM K-Hepes pH 7.7, 100 mM KCl, 1 mM $MgCl_2$, 5 mM EGTA, 0.1 mM $CaCl_2$, and 50 mM sucrose) containing protease inhibitors (10 μg/ml each of leupeptin, pepstatin and chymostatin). The eggs were then packed using a clinical centrifuge and crushed by centrifugation at 15000 g for 15 min at 4°C. The crude extract (the middle yellowish layer out of three layers) was collected, supplemented with protease inhibitors (10 μg/ml each of leupeptin, pepstatin and chymostatin) and 50 mM sucrose, snap-frozen in liquid $N_2$ as 10 μl aliquots and stored at −80∘C. Typically, for each extract batch a few hundred aliquots were made. Different extract batches exhibit similar behavior qualitatively, but the values of the contraction rate and disassembly rate vary (*Malik-Garbi et al., 2019*). All comparative analysis between conditions was done using the same batch of extract.

I-phase extract was prepared by adding $CaCl_2$ and cycloheximide to M-phase extract to concentrations of 0.4 μM and 2 μg/ml, respectively, and incubating at room temperature for a few minutes (*Field et al., 2011*; *Field et al., 2014*).

ActA-His was purified from strain JAT084 of Listeria monocytogenes (a gift from Julie Theriot, Stanford University) expressing a truncated actA gene encoding amino acids 1–613 with a COOH-terminal six-histidine tag replacing the transmembrane domain, as previously described (*Abu Shah and Keren, 2014*; *Abu-Shah et al., 2014*). Purified proteins were aliquoted, snap-frozen in liquid N2, and stored at −80°C until use.

ActA-His-Cys was purified from strain DP-L4363 of Listeria monocytogenes (a gift from Julie Theriot, Stanford University) and conjugated with ~6–8 molecules of Bodipy FL-X-SE (#D6102, Molecular Probes) per protein as previously reported (*Abu Shah and Keren, 2014*; *Abu-Shah et al., 2014*). ActA-Bodipy was stored at a concentration of ~30 μM at −80°C. Before use, ActA-Bodipy was sonicated on ice for 15 min and centrifuged at 4°C for 15 min at 16,000 g to remove aggregates.

Actin networks were labeled with GFP-Lifeact purified from transformed *E. coli* (gift from Chris Field, Harvard Medical School). The purified protein was concentrated to a final concertation of 252 µM in 100 mM KCL, 1 mM MgCl$_2$, 0.1 mM CaCl$_2$, 1 mM DTT and 10% Sucrose, and stored at −80°C until use.

## Emulsion preparation

An aqueous mix was prepared by mixing the following: 8 µl crude extract, 0.5 µl 20 × ATP regeneration mix (150 mM creatine phosphate, 20 mM ATP, 20 mM MgCl$_2$ and 20 mM EGTA) 0.5 µM GFP-Lifeact and any additional components as indicated. The final volume was adjusted to 10 µl by adding XB (10 mM Hepes, 5 mM EGTA, 100 mM KCl, 2 mM MgCl$_2$, 0.1 mM CaCl$_2$ at pH 7.8). The concentration of the components of the actin machinery in the mix can be estimated based on *Wühr et al. (2014)*. The total actin concentration is estimated to be ~20 µM. The ATP regeneration mix enables the system to continuously flow for more than 1–2 hr. Emulsions were made by adding ~3% by volume extract mix to degassed mineral oil (Sigma) containing 4% Cetyl PEG/PPG-10/1 Dimethicine (Abil EM90, Evonik Industries) and stirring for 1 min at 4°C. The emulsions were put in 30 µm or 100 µm thick chambers for imaging, or in 100 µm thick glass capillaries. 30 µm or 100 µm thick chambers were prepared by separating two passivated coverslips (*Abu-Shah et al., 2014*) with double sided tape, and sealing with VALAP (1:1:1 mix of vaseline, lanolin and paraffin). In 30 µm thick samples the imaged droplets were squished, allowing for better imaging of the actin network due to the flat glass-droplet surface, while in the 100 µm samples all but the largest droplets (R > 50 µm) were spherical, minimizing interactions with the interface and maximizing boundary symmetry.

Nocodazole treatment was performed by adding Nocodazole to a final concentration of 33 µM in the extract mix.

Bulk ActA assays were performed by adding Acta-His to the extract mix to the specified concentrations of 100 nM or 0.5 µM. Modulation of the interaction with the interface were performed by adding 25-100 nM of ActA-Bodipy to the extract mix.

## Magnetic manipulation

1 µm diameter Dynabeads MyOne Sterptavidin C1 superparamagnetic beads (Invitrogen) were washed 3 times and resuspended in XB to 200 µg/ml. The washed beads were incubated for 30 min with 0.6 µm biotin, washed 3 more times and resuspended in XB to 5 mg/ml (3–5 × 10$^6$ beads/µl). For magnetic manipulation assays, 0.5 µl magnetic beads (at 3 or 5 mg/ml) were added to the extract mix. For droplets in the size range examined (R ~ 50–100 µm) we estimate the number of beads in the aggregate to be 10–100 beads. Emulsions prepared as described above were loaded into rectangular glass capillaries (cross-section: 100 µm x 2000 µm) by capillary forces. The samples were incubated at room temperature for 10–15 min to allow the network to reach a steady state of contraction with a well-defined aggregate in the contraction center. Droplets in a symmetric configuration that were positioned near the capillary wall were identified and a horizontal magnetic force was applied by placing a magnetic needle (the tip of a steel sewing needle attached to a K and J Magnetics D14-N52 neodymium magnet 1/16' dia. x 1/4' thick), mounted on a three-axis micrometer manipulator, at a distance of 100–300 µm from the droplet (*Figure 3—figure supplement 1*). At this distance, the magnetic force on a single bead is ~0.1-2pN (depending on the distance of the magnetic needle). The magnetic needle was held in place for 40–120 s during which the contraction center moved toward the side of the droplet. Since the droplets are not anchored to the surface, the external force also leads to some movement of the entire droplet toward the capillary wall. The magnetic needle was then removed to allow the contraction center to reposition under the influence of internal forces. Bright-field and confocal images of the process were acquired using a spinning disk microscope as described below. The proximity to the capillary wall introduces some optical artifacts due to the glass curvature.

The magnetic force on a single bead was estimated by imaging beads in water in the capillaries and measuring their velocity toward the magnetic needle. Under the influence of the magnetic field, the beads formed straight chains that moved axially toward the magnet. Each chain was tracked and measured automatically using MATLAB. The drag force on each chain was estimated as the force on a cylinder, and the magnetic force per bead obtained by dividing by the estimated number of beads in the chain.

## Microscopy

Bright-field Images were acquired on a Zeiss Observer Z1 microscope using a Photometrics Cool-SNAP HQ2 CCD camera or a QuantEM camera. Confocal images were acquired using a Yokogawa CSU-X1 spinning disk attached to a Zeiss Observer Z1 microscope and acquired with an EM-CCD (QuantEM; Photometrics). Symmetry state statistics images of emulsions in 100 µm thick chambers were taken using 40x (NA = 1.3) or 63x (NA = 1.4) objectives. For each sample, 20–50 positions were chosen, containing droplets 30–120 µm in diameter, and z-stacks of bright-field images at a 3–5 µm separation of the sample were taken at several time-points between 15 to 60 min after sample preparation.

Magnetic manipulation experiments and aggregate tracking experiments were imaged by bright-field and spinning disk confocal microscopy in glass capillaries using a 20x (NA = 0.5) or a 10x (NA = 0.5) objective. Magnetic manipulation experiments were imaged at a single plane, with 2 s time interval for up to 15 min, starting several minutes after sample preparation. Aggregate tracking experiments were performed by following 3–5 droplets at a time, using 100 µm thick samples, 5–9 z-planes at 5 µm separation, at 30 s intervals. The imaging was initiated several minutes after sample preparation and lasted for 1 hr. Fluorescence spinning-disk images were taken using 488 nm and 561 nm lasers and appropriate emission filters. Device controls and image acquisition were carried out using Slidebook software.

## Analysis

Image analysis was carried out using custom-written code in Matlab. 2D and 3D positions of the droplets and aggregates were extracted from bright-field or confocal images and z-stacks, either manually or in an automated fashion. For the statistical analysis of symmetry states, the x-y positions of the droplet centers were determined automatically from the bright-field images using a circle detection built-in Matlab function based on the Circular Hough Transform. The x-y positions of the aggregate centers were determined by fitting a circle to the aggregates in the images manually. The z-position of the droplet and aggregate centers were determined manually from the z-stack. Three droplet size ranges – small (polar; denoted red in the figures), intermediate (transition; yellow) and large (symmetric; green) – were determined for each data set as follows: A threshold of $\frac{aggregate\ displacement}{droplet\ radius} = 0.4$ was used to categorize droplets as centered (0) or polar (1), and a symmetry breaking curve was defined as a moving Gaussian mean over droplet diameter (using a Gaussian window with a variance, $\sigma = 5\mu m^2$) of these binary symmetry states. A cumulative mean of the symmetry breaking curve was used to determine the border between the small (polar) and the intermediate droplet size ranges, where the cumulative mean drops below 0.95 of its max-min, and a reverse cumulative mean of 1 minus the symmetry breaking curve was similarly used to determine the border between the large (centered) and the intermediate droplet size ranges.

In the aggregate tracking experiments, the position of the contraction centers was determined from movies taken with 5–9 z planes at 5 µm separation at each time-point as follows. The initial droplet and aggregate positions were marked by hand. Subsequently in each frame the droplet position was determined by scanning the vicinity of the droplet position from the previous frame for a circle of the same size with maximum intensity. The vicinity of the aggregate from the previous frame was scanned for the brightest or highest-gradient elliptical ring in all z-planes, and the x-y position of the aggregate was defined as the centroid of this ring. The z-position was determined to sub-pixel resolution in the z-direction using a 3-point Gaussian interpolation over the ellipses scores (brightness or gradient) in the different z-planes. The z-position was determined as the center of the Gaussian.

Droplets that broke symmetry were defined as droplets which reached $\frac{aggregate\ displacement}{droplet\ radius} \geq 0.3$ during the movie. An initial estimate of the initiation time of the symmetry breaking process was defined to be the last time at which the displacement was 0.05 of the droplet radius (point A). The end of the symmetry breaking process in each droplet was determined as the first time the aggregate reached 0.97 of its maximum displacement (point C). The time of highest velocity-velocity correlation at 2 s interval between A and C was determined (point B), and the last point between A and B that had a negative radial velocity was taken as the start of the symmetry breaking process. If no such point exists, point A was considered instead. The duration of the symmetry breaking process was defined as the interval between the start and end points described above. The mean squared

displacement and velocity-auto correlation of the contraction centers of droplets that remained symmetric, and of the symmetry breaking process in droplets that broke symmetry, were analyzed from the tracks of the aggregates centers using MSDanalyzer MATLAB package described in *Tarantino et al. (2014)*.

The magnetic experiments were imaged in a single z-plane. The aggregate and droplet x-y positions were determined as detailed above. For analysis of the different recentering experiments as a function of time, the time in each experiment was defined by taking t = 0 to be the time at which the aggregate moved with zero velocity, or reached its minimal recorded velocity.

The actin network flow patterns were extracted by PIV analysis of time lapse movies of droplets as described previously (*Malik-Garbi et al., 2019*).

## Computational modeling

Mathematical modeling and numerical analysis of the Darcy flow, actomyosin network dynamics, forces on the network and aggregate, and aggregate positioning were done by solving Darcy equations, force balance equations and stochastic differential equations, as outlined in detail in the Appendices.

## Acknowledgements

We thank Yariv Kafri, Erez Braun, Guy Bunin and members of our lab for useful discussions and comments on the manuscript. We thank Nicolas Minc for comments on the manuscript, and Peter Lenart for useful discussions. We thank Gidi Ben Yoseph for excellent technical help.

This work was supported by a grant from the Israel Science Foundation (grant No. 957/15) to KK, a grant from the United States-Israel Binational Science Foundation (grant No. 2013275) to KK and A Mogilner, a grant from the United States-Israel Binational Science Foundation to KK and BG (grant No. 2017158), a grant by the US Army Research Office (grant W911NF-17-1-0417) to A Mogilner, and by grants from the Brandeis NSF MRSEC DMR-1420382 and the National Institutes of Health (R01-GM063691) to BG. A Manhart was partially supported by the National Institutes of Health (grant GM121971).

## Additional information

### Funding

| Funder | Grant reference number | Author |
|---|---|---|
| Israel Science Foundation | 957/15 | Kinneret Keren |
| United States-Israel Binational Science Foundation | 2013275 | Kinneret Keren<br>Alex Mogilner |
| United States-Israel Binational Science Foundation | 2017158 | Kinneret Keren<br>Bruce L Goode |
| Army Research Office | W911NF-17-1-0417 | Alex Mogilner |
| National Science Foundation | MRSEC DMR-1420382 | Bruce L Goode |
| National Institutes of Health | R01-GM063691 | Bruce L Goode |
| National Institutes of Health | GM121971 | Angelika Manhart |

The funders had no role in study design, data collection and interpretation, or the decision to submit the work for publication.

### Author contributions

Niv Ierushalmi, Angelika Manhart, Investigation, Writing - original draft, Writing - review and editing; Maya Malik-Garbi, Investigation, Writing - review and editing; Enas Abu Shah, Bruce L Goode, Resources, Writing - review and editing; Alex Mogilner, Kinneret Keren, Funding acquisition, Investigation, Writing - original draft, Writing - review and editing

## Author ORCIDs

Enas Abu Shah (iD) http://orcid.org/0000-0001-5033-8171
Bruce L Goode (iD) http://orcid.org/0000-0002-6443-5893
Alex Mogilner (iD) http://orcid.org/0000-0002-9310-3812
Kinneret Keren (iD) https://orcid.org/0000-0002-2962-6977

## Decision letter and Author response

Decision letter https://doi.org/10.7554/eLife.55368.sa1
Author response https://doi.org/10.7554/eLife.55368.sa2

## Additional files

### Supplementary files

- Transparent reporting form

### Data availability

All data generated or analysed during this study are included in the manuscript and supporting files. Custom code to conduct image analysis has been deposited to GitHub: https://github.com/nivieru/dropletsPositioning (copy archived at https://github.com/elifesciences-publications/dropletsPositioning).

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

## Appendix 1

## Centering Model

### Model Overview

Our model describes a two-phase system, consisting of an incompressible fluid phase and a viscous contracting actin network phase. Both phases coexist in a spherical droplet, with an impenetrable spherical aggregate located at the network contraction center. As an input to the simulation we use previously measured actin network dynamics (*Malik-Garbi et al., 2019*): the network contracts homogeneously toward the aggregate with a local flow velocity that is proportional to the distance from the aggregate, and turns over with a constant assembly rate and a disassembly rate that is proportional to the local network density (all model parameters are gathered in the tables below). We solve three coupled dynamic equations presented in detail below. The first one is the reaction-drift equation for the network density, which is essentially the mass conservation equation for the actin network describing its assembly, disassembly and flow. The second one is the Darcy flow equation to determine the flow velocities and pressure distributions in the cytosol. Finally, there is a force balance equation to determine the total force on the aggregate, or the aggregate velocity. Two type of simulations are performed. First, we perform a dynamic simulation in which the aggregate is free to move in response to the forces acting on it. To emulate the magnetic force pulling experiments, we allow the system to evolve to a symmetric steady state with the aggregate at the droplet center. We then translocate the aggregate to a decentered state, and use the shifted steady state network density distribution (with the network compressed to fit inside the droplet) as initial conditions for the simulation. From this point the aggregate is free to move according to the net forces on it determined by the network and fluid dynamics, and its position and velocity are measured. In the second type of simulation, the aggregate is held fixed in place, at a certain distance from the droplet center, and the system is allowed to reach a steady state in which the network dynamics, fluid motion, and forces are determined. As described below, we use this simulation to estimate the effective centering force on the aggregate.

### Model Description

All variables and parameters of the model and their values and orders of magnitude are provided in the tables below. We solve numerically the coupled system of equations for the actin density dynamics, the fluid dynamics and force balancing in a 3D geometry as described below. The equations are solved in a domain given by $\Omega := B_R(0) \backslash B_{r_A}(\mathcal{Q})$, where $B_R(0)$ is a sphere of radius $R>0$, and $B_{r_A}(\mathcal{Q})$ is an excluded spherical volume (the aggregate) of radius $r_A$ at position $\mathcal{Q}$.

  A. **Density dynamics**: We model the actin density $\rho(X,t)$ at position $X = (x,y,z)$ and time $t>0$ by the following equation:

$$\partial_t \rho + \nabla \cdot (V\rho) + \dot{\mathcal{Q}} \cdot \nabla \rho = \alpha - \beta\rho. \tag{A1}$$

  The first term on the l.h.s. is the rate of change of the density $\rho$, the second term describes the centripetal drift of the network with velocity $V(X,t)$ relative to the aggregate, and the third term accounts for the effective movement of the network with the aggregate's speed $\dot{\mathcal{Q}}(t)$. $\nabla$ denotes the gradient with respect to the spatial coordinates. The first and second terms on the r.h.s. are responsible for the actin assembly and disassembly with rates $\alpha$ and $\beta$, respectively. The boundary condition for *Equation A1* is zero density at the droplet boundary:

$$\rho(|X| = R, t) = 0. \tag{A2}$$

  Note that the Darcy friction forces have no influence on the actin network dynamics in this model. The justification for this is that, as we explain below, the magnitude of the Darcy

friction forces is in the ten pN range, while we assume that the myosin contraction and viscous forces in the actin network are at least an order of magnitude larger, based on previous estimates and measurements (**Malik-Garbi et al., 2019**).

B. **Fluid dynamics**: We solve numerically the Darcy flow equation for the fluid flow $U(X, t)$ and the pressure $p(X, t)$ (**Mogilner and Manhart, 2018**).

$$\frac{\kappa}{\mu} \nabla p = V + \dot{Q} - U, \tag{A3}$$

where $\nabla p$ is the local pressure gradient, $\mu$ is the fluid dynamic viscosity and $V(X, t)$ is the given actin network velocity relative to the aggregate (see **Equation A7** below). $\kappa$ is the network permeability coefficient, which is a function of network density $\rho$; we use the density-permeability relation given by:

$$\kappa = \frac{l^2}{\phi_f^{1/3}}, \quad l = \frac{1.87 \mu m \, \mu M^{1/2}}{\sqrt{\rho}}, \tag{A4}$$

where $\phi_f$ is the volume fraction of the fluid and $l$ is the network meshsize (**Charras et al., 2005**; **Schmidt et al., 1989**). The equation is complemented by the incompressibility condition for the fluid,

$$\nabla \cdot U = 0, \tag{A5}$$

and no-flux conditions at the droplet and aggregate boundary,

$$U(|X = R|) \cdot n = 0, \ U(|X - Q| = r_A) \cdot m = 0, \tag{A6}$$

where $n$ and $m$ are the local unit normal vectors at the droplet and aggregate boundaries, respectively. In **Equation A5**, we neglect the effect of variability of the small volume fraction of actin. Based on measurements, we assume that the actin centripetal contraction speed relative to the aggregate $V$ increases linearly away from the aggregate and has the following form:

$$V = \gamma(Q - X) \left[ 1 - \frac{r_A}{|Q - X|} \right]. \tag{A7}$$

The term in the square brackets accounts for the observation that the actin flow becomes zero at the aggregate boundary.

## Model of recentering dynamics

In order to find the aggregate speed $\dot{Q}$ at each time step, we use the assumption that the total Darcy friction force on the whole actin network is equal to zero (more precisely, that forces other than the total Darcy friction force are negligibly small):

$$\int_\Omega \frac{\mu}{\kappa} (V + \dot{Q} - U) d\Omega = 0, \tag{A8}$$

where $d\Omega$ is the volume element. The aggregate speed is determined by an iteration procedure from **Equation A3** at each computational step.

Initial Density Condition: To mimic the experimental setup in the magnetic pertubation experiments, we performed two steps to calculate the initial actin network density:

1. We assumed $Q = \dot{Q} = 0$, i.e. the aggregate is positioned at the droplet center. We simulated **Equations A1 and A2** on a time interval sufficiently large to attain a stationary density distribution.
2. Then the aggregate and actin density were shifted together away from the center along the x-axis toward the droplet boundary, and the actin on the side closer to the droplet edge was compressed to remain inside the droplet. The resulting density was used as initial condition

for further simulations to determine the dynamics of the actin network, the fluid, and the position of the aggregate.

## Model with a fixed contraction center

In this case, we shifted the aggregate to position $\mathcal{Q}$ from the aggregate center and fixed the aggregate at this position. Consequently we used $\dot{\mathcal{Q}} = 0$ in **Equations A1 and A3**. We first solved **Equations A1-A2** on a time interval sufficiently large to attain a stationary density distribution. Then we solved **Equations A1, A4 and A5** to obtain the pressure and fluid flow. The main change was in **Equations A8**: Now that the aggregate is held in place, there is a net total Darcy friction force applied to it given by the equation

$$F = \int_{\Omega} \frac{\mu}{\kappa}(V - U)\,\mathrm{d}\Omega. \tag{A9}$$

We used this equation to calculate the effective hydrodynamic centering force on the aggregate and varied the aggregate position and droplet radius to find the force dependence on these parameters.

## Simulation

### Reformulation in cylindrical coordinates

Note that all above equations are given in 3D-space. However we have cylindrical symmetry along the axis on which the aggregate moves (the $x$-axis): If positioned at $\mathcal{Q} = (Q, 0, 0)$, the aggregate will continue to move along the $x$-axis, i.e. $\dot{\mathcal{Q}} = (\dot{Q}, 0, 0)$. Switching to cylindrical coordinates $(x, y, z) \rightarrow (x, r, \varphi)$,

$$x = x, \quad y = r\cos\varphi, \quad z = r\sin\varphi, \tag{A10}$$

we can replace **Equations A3, A4 and A5** by equations for $\tilde{p}(x, r)$ and $\tilde{U}(x, r) = (U_x(x, r), U_r(x, r))$, where $U_x$ and $U_r$ are the fluid components of the solution in the $z = 0$ plane in the $x$ and in the radial direction, respectively.

$$\frac{\kappa}{\mu}\nabla\tilde{p} = \tilde{V} + \dot{\tilde{Q}} - \tilde{U}. \tag{A11}$$

$$\tilde{V} = \gamma \begin{pmatrix} Q - x \\ -r \end{pmatrix}\left[1 - \frac{r_A}{|Q - X|}\right], \quad \tilde{Q} = \begin{pmatrix} Q \\ 0 \end{pmatrix} \tag{A12}$$

Note that now $\nabla = (\partial_x, \partial_r)$. The main change can be seen in the incompressibility condition which becomes,

$$\nabla \cdot \tilde{U} = -\frac{1}{r}U_r, \tag{A13}$$

which is a consequence of the use of cylindrical coordinates. The boundary conditions are unchanged. Note that $\tilde{U}$ and $\tilde{p}$ are the solution in the $z = 0$ plane. The full solution can then be obtained for the pressure from $p(x, y, z) = \tilde{p}(x, r)$, and for the fluid speed by rotation, i.e. $U(x, y, z) = (U_x, \cos\varphi\, U_r + \sin\varphi\, U_r)$, where $r = \sqrt{y^2 + z^2}$.

We can also rewrite the actin density equation in cylindrical coordinates, which yields an equation for $\tilde{\rho}(x, r, t)$:

$$\partial_t\tilde{\rho} + \nabla \cdot (\tilde{V}\tilde{\rho}) + \dot{\tilde{Q}} \cdot \nabla\tilde{\rho} = \alpha - \beta\tilde{\rho}. \tag{A14}$$

Again we can obtain the full solution by $\rho(x, y, z, t) = \tilde{\rho}(x, r, t)$.

## Simulation details

We used the Lax-Friedrichs scheme for the density equation on a rectangular grid and a Finite Element Method for the fluid simulation using Matlab's PDE toolbox. At each time interval $[t, t + dt]$ we performed the following steps:

1. Determine aggregate speed such that (*Equations A8*) is fulfilled by repeatedly solving (*Equations A11*), (*Equations A13*).
2. Calculate the fluid flow from (*Equations A11*), (*Equations A13*) using the correct aggregate speed.
3. Solve the density *Equations A14* on $[t, t + dt]$ using the correct aggregate speed.

The density was interpolated from the rectangular grid to the triangular FEM grid at each time step. To solve the fluid equation we reformulated (*Equations A11*), (*Equations A13*) in terms of the pressure only. This can be done by taking the divergence of (*Equations A11*) and using (*Equations A13*). Tables *Equations A1, A2 and A3* list the used parameters and variables, unless specified otherwise.

## Parameters

### Appendix 1—table 1. Variables.

**Variables**

| Name | Meaning | Computed order of magnitude |
|---|---|---|
| $U$ | Fluid velocity | 10µm/min |
| $V$ | Actin network velocity | 10µm/min |
| $\rho$ | Actin network density | 5µM |
| $f$ | Hydrodynamic force density | fN/µm³ |
| $Q$ | Distance of the aggregate to the droplet center | 10µm |

### Appendix 1—table 2. Parameters.

**Biological parameters**

| Name | Meaning | Value | Reference |
|---|---|---|---|
| $\mu$ | Fluid viscosity | 1.7 10⁻⁴pN min/µm² | 10x the viscosity of water |
| $\kappa$ | Actin network permeability | see (*Equation A4*) | Equation (19) in **Schmidt et al. (1989)** and in Supp Mat of **Charras et al. (2005)** |
| $\gamma$ | Contraction rate | 0.67/min (M-phase) 0.26/min (I-phase) | Measured experimentally (**Malik-Garbi et al., 2019**). |
| $\alpha$ | Actin assembly rate | 1 µM/min | Estimated to give experimentally measured density |
| $\beta$ | Actin disassembly rate | 1.43/min | Measured experimentally (**Malik-Garbi et al., 2019**). |
| $R$ | Droplet radius | 20–60 µm | Measured experimentally |
| $r_A$ | Aggregate radius | 4–12 µm | Proportional to droplet radius, measured experimentally (**Malik-Garbi et al., 2019**). |

**Appendix 1—table 3. Numerical Parameters.**

**Numerical parameters**

| Name | Meaning | Value |
|---|---|---|
| $\Delta x$ | Spatial step, density simulation | 0.5–0.75 μm |
| $\Delta t$ | Temporal step, density simulation | $\approx 10^{-3}$ min |
| $dt$ | Temporal step | 0.5 min |
| $H_{\max}$ | Maximum mesh edge length of FEM mesh | 1 μm |

## Appendix 2

### Symmetry Breaking Model

The symmetry breaking is modeled as driven by a stochastic two-state interaction with the boundary. When the interaction is *on*, the aggregate moves outward in the radial direction at a constant velocity, and when the interaction is *off*, the aggregate recenters under the influence of the centering force discussed above. Additionally, the aggregate undergoes a small random Brownian motion. The movement is modeled in 3D, but the interaction with the boundary is always taken to be in the current radial direction (i.e. with the closest interface).

### Model Parameters

The symmetry breaking velocity, $V_{out}$, is estimated from the experiments and taken as a constant for simplicity. The centering velocity $V_{cen}$ is size-independent and linear in the displacement from the droplet's center as observed in the experiments and in the centering simulation, $V_{cen} = K_{in} \cdot d$, where $d$ is the distance from the center. The rate of transition to the *on* state, $k_{on}$, is taken to be inversely proportional to the distance of the aggregate from the droplet edge, $k_{on} \propto \frac{1}{R-d}$, and its order of magnitude is estimated from the experiments. This dependence accounts in a simplified manner for the increase in the probability of the network to interact with the boundary, as the network density near the edge increases as the aggregate moves closer to the boundary. Similarly $k_{off}$, the transition rate to the *off* state, is proportional to that distance, $k_{off} \propto R - d$. For both rates, the proportionality factor is varied manually within the same order of magnitude to simulate different interaction strengths. Thus we have:

$$k_{off} = r_{off} \times (R - d_i)$$

$$k_{on} = r_{on} \times \frac{1}{(R - d_i)}$$

where $r_{on}$ and $r_{off}$ are constant. Finally, the aggregate diffusion coefficient $D$ is estimated from the experiments, based on the random fluctuations of the aggregate in the centered state (***Figure 2d***).

### Simulation

A collection of droplets of various sizes is initiated in a centered state. At each time step $i$, a random variable for each droplet determines whether the interaction will switch state or remain the same, based on the relevant on and off rates. The aggregate moves outward or inward based on the interaction state, with an additional random diffusive motion. If the aggregate reaches displacement $d \geq 0.75R$, it remains there for the rest of the simulation.

We model the stochastic trajectory of the aggregate by a combination of dynamical Markov process method and Langevin equation. We use a random number generator and dynamical Markov process method to simulate the switches between the on and off states as follows. Let $rr \in [0, 1]$ be a uniformly distributed random variable, if the state $s_i$ is *off* and $rr < k_{on} \cdot \Delta t$, the system transitions to state $s_{i+1}$ *on*. If the state $s_i$ is *on* and $rr < k_{off} \cdot \Delta t$, then the system transitions to state $s_{i+1}$ *off*. Then the subsequent movement is dependent on the interaction on/off state. If $s_{i+1}$ is on (i.e. the interaction with the droplet boundary is engaged), we solve the following Langevin equation:

$$\vec{d}_{i+1} = \vec{d}_i + V_{out} \cdot \hat{d}_i \Delta t + \sqrt{2D\Delta t} \cdot \vec{\eta}$$

otherwise

$$\vec{d}_{i+1} = \vec{d}_i - K_{in} \cdot \vec{d}_i \Delta t + \sqrt{2D\Delta t} \cdot \vec{\eta}$$

where $\vec{\eta}$ is a standard normal distributed random vector.

**Appendix 2—table 1.** Parameters.

**Parameters**

| Name | Meaning | Value | Reference |
|------|---------|-------|-----------|
| $V_{out}$ | Symmetry breaking velocity | 2 μm min$^{-1}$ | Measured experimentally (**Figure 2b**) |
| $K_{in}$ | Centering velocity coefficient | 0.5 min$^{-1}$ | Measured experimentally (**Figure 3g**) and from simulation |
| $r_{off}$ | Off-rate coefficient | 0.0013- 0.0125 μm$^{-1}$ min$^{-1}$ | Order of magnitude estimated from experiment (**Figure 2b**) |
| $r_{on}$ | On-rate coefficient | 4 - 40 μm min$^{-1}$ | Estimated to give reasonable results |
| $D$ | Aggregate diffusion coefficient | 0.5 μm$^2$ min$^{-1}$ | Measured experimentally |
| $R$ | Droplet radius | 15 - 100 μm | |
| $\Delta t$ | Time step | 0.1 min | |

