## [Decision Letter]

**Acceptance summary:**

This manuscript reveals biophysical principles of centering organelles in large cells. The mechanism is based on a cell-wide contractile F-actin network, which centers objects by a hydrodynamic mechanism between the network and the cytoplasm, and is independent of viscosity and network dynamics. These results have key relevance for understanding how the cytoplasm of mouse oocytes is organized, which then sets the polarity axes of the early embryo.

**Decision letter after peer review:**

Thank you for submitting your work entitled "Centering and symmetry breaking in confined contracting actomyosin networks" for consideration by *eLife*. Your article was reviewed by a Senior Editor, and guest Reviewing Editor, and two reviewers. The reviewers have opted to remain anonymous.

Unfortunately, the reviewers found the experiments elegant and the physical model interesting, but in the end, did not feel that the model is proven by the data provided. They were concerned that the story was weak in connecting the observation and model to the behavior of any real cell, and that the physical model is not backed up by rigorous experimental testing. In the absence of a link to an actual cellular phenomenon, they were unable to support presentation in *eLife*.

Reviewer #1:

The manuscript by Ierushalmi et al., proposes a new mechanism for centering and positioning inside large cells by actomyosin networks. The authors develop a nice in vitro system in which *Xenopus* mitotic extract are contained within oil-droplets. In these conditions, self-organized contractile actomyosin structures are generated. The authors propose that this system could reveal novel function for these cytoskeletal structures. Overall, I find the paper interesting, as it offers a nice experimental system amenable to manipulation (for example the use of magnetic beads to study centering forces is insightful). Also, the observation that actomyosin network can generate either a polarized or a centered state depending on droplet size is intriguing. Finally, the observations that centering could be the result of Darcy friction and polarization the result of interactions between actomyosin and the cortex are interesting. My enthusiasm for this paper is however tempered by (1) The limited experimental data to support the proposed centering mechanism, (2) The lack of discussion of how this mechanism could interface with cell cycle signals, which dominate the early stage of embryogenesis and have already been shown to be essentially for positioning of subcellular structures, most notably nuclei.

An important theoretical result of this paper is that the observed centering mechanism can be physically explained by poroelastic effects. It is proposed theoretically that the relative motion of the contractile actomyosin network and the cytosol generates forces in the correct direction for centering by Darcy friction. This is a very interesting result. However, there is very little experimental effort to verify the theoretical predictions. It should be possible in this system to measure actomyosin flow and flow of the cytosol and show that relative motion of the order predicted by the theory is observed, etc. I do not believe that the data presented in Figure 4G-H are a strong validation of the theory.

Another significant limitation of this paper is that it fails to acknowledge the importance of cell cycle signaling in the centering processes of oocytes and embryos. All the discussed examples in the Introduction, including ooplasmic segregation in zebrafish, are coordinated in space and time by oscillations in biochemical activities linked to the cell cycle. This is because bulk actomyosin contractility is controlled by the cell cycle. The authors should discuss that and cite at least this classic paper (Field et al., 2011). The coupling of cell cycle signals by mitotic phosphatase PP1 and cytoskeletal forces in positioning nuclei in *Drosophila* embryos has recently been dissected using sophisticated imaging methods (Deneke et al., 2019). My assumption is that in the authors' system the cytoplasmic extracts stay in mitosis. Notice that mitosis is a transient state and one also characterized by significant other cytoskeletal rearrangements. It seems important that the authors discuss how their mechanism could interface with spatial cell cycle signaling. For example, if they repeated their experiments with interphase extracts would they see similar phenomena? My guess is that would not happen, based on the paper above by Field et al. These would be important things to know to determine how relevant the findings of this paper could be. Otherwise, the in vivo relevance of this study for early embryogenesis – which is dominated by rapid cleavage cycles- as well as other cellular stages remains unclear.

Reviewer #2:

This manuscript uses an elegant combination of biophysics experiments and modeling to address question of how cytoplasmic flows drive centered or polarized location of subcellular compartments. This is an important cell biological question, especially in oocytes, and the authors have designed elegant experiments to demonstrate a putative mechanism involving the fluid dynamics surrounding a contractile actin meshwork in an artificial cell constructed by encapsulating *Xenopus* extracts in emulsion drops of varying size. In recent work the authors found that the extract spontaneously forms a steady state contractile flow around a central contraction "aggregate" center. Here they demonstrate that dynamic steady state can either act as a centering mechanism or spontaneously break symmetry and drive polarized motion of the aggregate towards the periphery, depending on the droplet size. They then build a physical model that captures this behavior, taking into consideration the fluid dynamics induced by the contracting mesh work and the boundary conditions of the contractile system. The strengths of this manuscript are the elegance of the experimental and theoretical biophysical approaches. The rigor with which the authors demonstrate a putative mechanism by which centering can be regulated by cell size and with dynamic contractile networks is top notch. The extent to which we know of biological systems which harness this mechanisms is not as clear, and unfortunately, this detracted from my appreciation of this work presented.

Essential revisions:

-In Figure 5, the authors show how modifications to the attractive interactions with the cell boundary can alter the tendency for a polarized phenotype. The comparison between the experimental and theory data is lovely, and I would love to see other perturbations/control parameters explored using this similar approach, such as the mesh size and fluid viscosity to test the recentering model. Some of this data is put in Figure 4—figure supplement 4, but it is not clear to me what the authors mean by how increased actin disassembly "enhances contraction". I think the link between how biochemical factors can be used to control the emergent centering/polarization of the system could be made stronger in the main figures.

- Due to its role in modifying actin assembly dynamics, I have concern with the use of ActA in Figure 5. Did the authors try other means to modify boundary attachments? Is there evidence that using ActA alters the mesh size of the contractile mesh?

- I'm curious to how the changes to the parameters (experimentally or in the model) affect the observed centering force and kinetics.

- The extracts in emulsion system is fairly artificial. I would be interested for the authors to speculate a bit more about biological situations where this general principle may be useful. The authors mention migratory cells, but it is not clear to me how important the Darcy forces will be when considering contraction of a 2D mesh? Or, are the situations where connections to the boundary are known to drive centering vs. polarization?

[Editors' note: the decision letter after the authors submitted for reconsideration follows.]

Thank you for submitting your article "Centering and symmetry breaking in confined contracting actomyosin networks" for consideration by *eLife*. Your article has been reviewed by Suzanne Pfeffer as the Senior Editor and Reviewing Editor, and three reviewers. The reviewers have opted to remain anonymous.

The reviewers have discussed the reviews with one another and the Reviewing Editor has drafted this decision to help you prepare a revised submission.

In this paper, Ierushalmi et al., propose a novel hydrodynamic mechanism for the positioning of cellular components. The authors use *Xenopus* extracts encapsulated in emulsion droplets, where continuous actomyosin activity accumulates extract components into an aggregate. They find that the positioning of this aggregate is a function of the droplet size with a polar localization in smaller droplets and a centered localization in larger droplets. To understand the origin of the aggregate positioning, the authors develop a model based on the interplay between actomyosin generated flows and Darcy friction with the surrounding fluid. The authors find quantitative agreement between the model and the measured recentering dynamics in both interphase and meiotic extracts, and a similar transition range between polar and centered aggregates as a function of droplet radii. Importantly, the proposed centering mechanism is generated by internal flows and is independent of direct interactions with the droplet's surface.

The reviewers would like a bit more discussion of the following points and ask if it might be possible to present one additional comparison (but this is not essential) as follows:

*Reviewer 1:*

I think that it is very unlikely that nuclear positioning in *Xenopus* embryos would be controlled by the mechanisms elucidated in this paper. Nuclei are not centered in meiosis and while they are at fertilization that process is likely to be microtubule-dependent. Moreover, centering happens in interphase when the proposed mechanism is weaker due to the reduced contractility of bulk actomyosin, while microtubules grow rapidly when Cdk1 activity is low. The story might be different for other organelles and I would not be surprised if the authors' model turns out to be important. Please discuss.

*Reviewer 2:*

I agree that it is rather unlikely that this mechanism is at work in *Xenopus oocytes* or zygotes, simply due to the dimensions and how yolk and other organelles are organized in these cells. However, these observations in *Xenopus* extracts reminded me very much to mammalian (mouse) oocytes. While quite some work has been done directly in mouse, also on the biophysical side (e.g. by the Verlhac/Terret laboratory), I am not aware of a study that would provide a similarly detailed mechanistic model, and comparably rigorous test of the model. Therefore, the present study will be inspirational/instructive for groups working on mammalian oocytes and early embryos. I also agree that addition of different cell cycle states was critical. In the current manuscript, the weakest link is that de-centering/symmetry breaking is major point even mentioned in the title, but in fact the mechanism of centering is what really is worked out in detail. The model for de-centering is rather coarse, and I would be happy to see more discussion on this even if it is a bit on the speculative side.

*Reviewer 3:*

The only quantitative comparison between experiments and the model are the displacement vs. time curves for the aggregate recentering dynamics (e.g. Figure 4). It would nice to include an additional comparison. For example, the authors could include a plot of the recentering velocity vs displacement curves like in Figure 3G that compares experiments and simulations for the I-phase and M-phase extracts.

---

## [Author Response]

Both reviewers found the experiments elegant and the modeling interesting. We thank the reviewers for their thoughtful and insightful comments. As detailed in the point-by-point response to the reviewers below we can address nearly all of the issues raised by the reviewers. In particular, we have made a considerable effort, both experimentally and theoretically, to provide additional support for the proposed centering mechanism and explore how it could interface with cell cycle signals.Reviewer #1:

*The manuscript by Ierushalmi* et al.*, proposes a new mechanism for centering and positioning inside large cells by actomyosin networks. The authors develop a nice* in vitro *system in which Xenopus mitotic extract are contained within oil-droplets. In these conditions, self-organized contractile actomyosin structures are generated. The authors propose that this system could reveal novel function for these cytoskeletal structures. Overall, I find the paper interesting, as it offers a nice experimental system amenable to manipulation (for example the use of magnetic beads to study centering forces is insightful). Also, the observation that actomyosin network can generate either a polarized or a centered state depending on droplet size is intriguing. Finally, the observations that centering could be the result of Darcy friction and polarization the result of interactions between actomyosin and the cortex are interesting.*

We thank the reviewer for the appreciation of our work.

My enthusiasm for this paper is however tempered by (1) The limited experimental data to support the proposed centering mechanism, (2) The lack of discussion of how this mechanism could interface with cell cycle signals, which dominate the early stage of embryogenesis and have already been shown to be essentially for positioning of subcellular structures, most notably nuclei.

As detailed below, we have undertaken a considerable theoretical and experimental effort in response to the reviewer’s comments, to provide additional support for the proposed centering mechanism and explore how it could interface with cell cycle signals.

An important theoretical result of this paper is that the observed centering mechanism can be physically explained by poroelastic effects. It is proposed theoretically that the relative motion of the contractile actomyosin network and the cytosol generates forces in the correct direction for centering by Darcy friction. This is a very interesting result. However, there is very little experimental effort to verify the theoretical predictions. It should be possible in this system to measure actomyosin flow and flow of the cytosol and show that relative motion of the order predicted by the theory is observed, etc. I do not believe that the data presented in Figure 4G-H are a strong validation of the theory.

The hydrodynamic mechanism arises from Darcy friction forces due to the relative motion of the contractile actomyosin network and the cytosol as noted by the reviewer. We provide measurements of the actin network (Figure 1C). Since the fluid is incompressible, and the friction with the network is relatively small, the contractile network flow is not expected to generate appreciable fluid flow in the bulk cytosol (note the different scales in Figure 4B and 4C), so the relative velocity of the network relative to the cytosol is primarily due to the network velocity (In fact simulations show the centering mechanism works even if the fluid remains stationary). While we agree with the reviewer that mapping the fluid flow in the system would be informative, measuring such small fluid flows (<5μm/min; except very close to the droplet’s interface) in the presence of a contracting network is difficult, since tracer particles that are large enough to diffuse sufficiently slow to facilitate tracking tend to get trapped in the network.

We do however provide additional validation for the centering mechanism in the revised manuscript in a different manner, by varying the properties of the actomyosin network and showing that the model (with no fit parameters) is able to predict the dependence of the centering dynamics on the system’s properties. Theoretically, we added model simulations showing that the centering mechanism is robust to changes in the viscosity of the network and the actin meshwork assembly/disassembly, but is highly dependent on the rate of actin network contraction (Figure 4F, Figure 4—figure supplement 1). We have performed additional experiments, characterizing the centering process under different conditions and showed that the model simulations are able to quantitatively predict the centering dynamics in the different experimental conditions with no fit parameters. Specifically, we have added experiments where we change the cell cycle state of the system (Figure 4I-K) or add nucleation promoting factors (Figure 4—figure supplement 2). We follow the centering dynamics under these different conditions and show that the model predicted results recapitulates the observed relation between the centering dynamics and the network contraction rate (Figure 4I-K, Figure 4—figure supplement 2).

*Another significant limitation of this paper is that it fails to acknowledge the importance of cell cycle signaling in the centering processes of oocytes and embryos. All the discussed examples in the Introduction, including ooplasmic segregation in zebrafish, are coordinated in space and time by oscillations in biochemical activities linked to the cell cycle. This is because bulk actomyosin contractility is controlled by the cell cycle. The authors should discuss that and cite at least this classic paper (Field* et al., *2011). The coupling of cell cycle signals by mitotic phosphatase PP1 and cytoskeletal forces in positioning nuclei in Drosophila embryos has recently been dissected using sophisticated imaging methods (Deneke* et al., *2019). My assumption is that in the authors' system the cytoplasmic extracts stay in mitosis. Notice that mitosis is a transient state and one also characterized by significant other cytoskeletal rearrangements. It seems important that the authors discuss how their mechanism could interface with spatial cell cycle signaling. For example, if they repeated their experiments with interphase extracts would they see similar phenomena? My guess is that would not happen, based on the paper above by Field* et al. *These would be important things to know to determine how relevant the findings of this paper could be. Otherwise, the* in vivo *relevance of this study for early embryogenesis – which is dominated by rapid cleavage cycles- as well as other cellular stages remains unclear.*

We thank the reviewer for raising this important point, and have added experiments to directly address this point in the revised manuscript. Indeed, it has been demonstrated in many systems that the cell cycle induces dramatic changes in cytoskeletal dynamics as well as functionally important alterations of the subcellular localization of various cellular components. The paper by Field et al., that the reviewer mentions is a nice example of the significant cell-cycle induced changes in the properties of the actomyosin cytoskeleton in early embryonic development in *Xenopus*. As the reviewer notes, this is relevant to our current study and we now cite this paper in the revised manuscript. Following the reviewer’s comment, we have also experimentally examined how changes in the cell cycle influence the centering mechanism we found. The *Xenopus* extracts we use are arrested in metaphase of meiosis II, and it is possible to cycle them into interphase by adding Ca ions. As expected from the results by Field et al., we observe a sparser actin network which exhibits much weaker contraction with the I-phase extracts. Our results show that the substantially reduced actomyosin contractility in I-phase extracts weakens the centering force significantly, resulting in more symmetry breaking (Figures 4I-K, Figure 4—figure supplement 3). These results illustrate how the effect of the hydrodynamic centering mechanism could interface with changes in the cell cycle state of the cell, supporting the biological relevance of this mechanism as a biologically tunable positioning mechanism.

Reviewer #2:This manuscript uses an elegant combination of biophysics experiments and modeling to address question of how cytoplasmic flows drive centered or polarized location of subcellular compartments. This is an important cell biological question, especially in oocytes, and the authors have designed elegant experiments to demonstrate a putative mechanism involving the fluid dynamics surrounding a contractile actin meshwork in an artificial cell constructed by encapsulating *Xenopus* extracts in emulsion drops of varying size. In recent work the authors found that the extract spontaneously forms a steady state contractile flow around a central contraction "aggregate" center. Here they demonstrate that dynamic steady state can either act as a centering mechanism or spontaneously break symmetry and drive polarized motion of the aggregate towards the periphery, depending on the droplet size. They then build a physical model that captures this behavior, taking into consideration the fluid dynamics induced by the contracting mesh work and the boundary conditions of the contractile system. The strengths of this manuscript are the elegance of the experimental and theoretical biophysical approaches. The rigor with which the authors demonstrate a putative mechanism by which centering can be regulated by cell size and with dynamic contractile networks is top notch.

We thank the reviewer for the appreciation of the elegance and rigor of our work.

The extent to which we know of biological systems which harness this mechanisms is not as clear, and unfortunately, this detracted from my appreciation of this work presented.

To enhance the relevance of our manuscript for biological systems, we have added experiments in which we change the cell cycle state of the system. We now show how the cell-cycle induced change in the actomyosin contraction induces a large change in the centering dynamics and aggregate localization. We find that the hydrodynamic forces are sufficient to induce centering even when the network flow is reduced several fold, but the centering process is much slower.

Since the system we use has physiological characteristics, and in particular the actin network flow rates we observe are comparable to those measured in cells, our results indicate that the hydrodynamic mechanism that induces centering in our system will generate similar forces in vivo. Obviously, the influence of these hydrodynamic forces on the localization of sub-cellular components within living cells will depend on all the other factors in the system, and need not be as simple as the centering we observe in our artificial cells. Nevertheless, we believe our system highlights the influence of hydrodynamic forces in the cytosol and provides potential localization mechanism which are relevant for understanding the localization mechanisms in living cells, an in particular large cells such as oocytes.

Essential revisions:-In Figure 5, the authors show how modifications to the attractive interactions with the cell boundary can alter the tendency for a polarized phenotype. The comparison between the experimental and theory data is lovely, and I would love to see other perturbations/control parameters explored using this similar approach, such as the mesh size and fluid viscosity to test the recentering model. Some of this data is put in Figure 4—figure supplement 4, but it is not clear to me what the authors mean by how increased actin disassembly "enhances contraction". I think the link between how biochemical factors can be used to control the emergent centering/polarization of the system could be made stronger in the main figures.

We thank the reviewer for this excellent suggestion, and have undertaken a considerable experimental and theoretical effort to address this point and enhance the comparison between model and experiment by studying additional perturbations. Theoretically, we have studied the effect of changing the different model parameters on the centering process. We find that there is essentially a single control parameter for the centering process, namely the network contraction rate. Interestingly, as explained in the revised manuscript, changing the fluid viscosity or the actin meshwork assembly and disassembly dynamics which control the meshwork permeability has little effect on the centering process, since these parameters influence both the hydrodynamic force driving the centering process and the resisting drag in a similar manner. In contrast, the network contraction rate mainly influences the hydrodynamic centering force and is thus predicted to have a strong influence on the centering dynamics. The dependence of the centering force on the network contraction rate is now shown in Figure 4F, and the results of varying all the different model parameter are shown in Figure 4—figure supplement 1.

Experimentally, we found different perturbations that lead to networks that contract at different rates. Specifically, changing the cell cycle state of the system to interphase reduces actomyosin contractility and slows down contraction (as previously shown by Field et al., 2011) and adding a nucleation promoting factor for branched actin structures which makes the network more rigid also results in slower contraction. Importantly, we show that by incorporating the measured actin network contraction rates for each condition, the model is able to quantitatively predict the changes in the centering dynamics with no fit parameters. We believe these added results provide important insight, showing that the network contraction rate (which is an easily observable quantity, both in vitro and in vivo) is the only important control parameter, and providing additional corroboration for the model. As suggested by the reviewer, we include these additional results in the main figures (Figure 4F,I-K) as well as in Figure 4—figure supplement 1 and Figure 4—figure supplement 2.

- Due to its role in modifying actin assembly dynamics, I have concern with the use of ActA in Figure 5. Did the authors try other means to modify boundary attachments? Is there evidence that using ActA alters the mesh size of the contractile mesh?

In order to address this point, we have done additional control experiments with the same concentrations of cytosolic ActA. The relatively low amount of cytosolic ActA (100nM) does not have a large influence on the localization pattern, whereas the same concentration of cortexbound ActA-bodipy leads to complete symmetry breaking. We believe that this result shows that the important factor for symmetry breaking in Figure 5 is the enhanced attachment of actin to the interface, as opposed to changes to the network structure promoted the relatively low concentration of ActA. These results are shown in Figure 4—figure supplement 4 in the revised manuscript.

- I'm curious to how the changes to the parameters (experimentally or in the model) affect the observed centering force and kinetics.

We thank the reviewer for suggesting this. As explained above, we performed additional simulations and experiments under different conditions to address this point directly. The simulations show that while the centering forces depend on many parameters, the centering kinetics effectively depend only on the network contraction rate (Figure 4—figure supplement 1). Moreover, as explained above, we show that the same model is able to quantitatively account for the observed centering dynamics, for two different perturbations in which the network dynamics exhibit significant changes in the network contraction rates (Figures 4I-K, Figure 4—figure supplement 2).

- The extracts in emulsion system is fairly artificial. I would be interested for the authors to speculate a bit more about biological situations where this general principle may be useful. The authors mention migratory cells, but it is not clear to me how important the Darcy forces will be when considering contraction of a 2D mesh? Or, are the situations where connections to the boundary are known to drive centering vs. polarization?

In 2D, Darcy forces would probably be negligible. However, in 3D, our in vitro study actually involved sizes, densities and speeds within physiological regimes, characteristic of large cells. It was demonstrated before that cytoplasmic flows play a role in positioning organelles in plant cells (Verchot-Lubicz, 2010, Chebli, 2013), *C. elegans* one-cell embryo (Niwayama, 2012). Hydrodynamic forces and flows seem to contribute to positioning of multiple nuclei in early myotubes (Wilson, 2012). More specifically, 3D cytoplasmic flows generated by active deformations of actin networks and Darcy friction between these deforming networks and viscous fluids were shown to be responsible for a) spindle decentering in oocytes (Yi 2011), and b) yolk granules positioning in oocytes (Shamipour, 2019). Last, but not least, positioning of nucleus in 3D cell migration is of fundamental importance (Zhu, 2018), and hydrostatic/hydrodynamic effects do contribute significantly not only to this positioning, but in fact to the process of migration itself (Petrie, 2017, Zhu, 2018). These examples (the number of which is growing) suggest that cytoplasmic flows and pressure gradients, generated by either contracting actomyosin networks, or microtubule/dynein/kinesin networks, or both, do contribute in some physiologically important cases to organelle positioning and cell deformation and movements. Detailed biophysical investigation of these in vivo systems is very difficult. Our system hence provides a useful and novel in vitro model for these in vivo systems.

[Editors' note: the decision letter after the authors submitted for reconsideration follows.]

Reviewer 1:I think that it is very unlikely that nuclear positioning in *Xenopus* embryos would be controlled by the mechanisms elucidated in this paper. Nuclei are not centered in meiosis and while they are at fertilization that process is likely to be microtubule-dependent. Moreover, centering happens in interphase when the proposed mechanism is weaker due to the reduced contractility of bulk actomyosin, while microtubules grow rapidly when Cdk1 activity is low. The story might be different for other organelles and I would not be surprised if the authors' model turns out to be important. Please discuss.

We have revised the manuscript to discuss the possible biological relevance of this mechanism. In the discussion of the revised manuscript, we now explain that the proposed mechanism is expected to be relevant in cases where localization is actin-dependent and large-scale actin network flows are present, and note that it is unlikely to be relevant for nuclear positioning in *Xenopus* embryos which is primarily microtubule-dependent.

Reviewer 2:I agree that it is rather unlikely that this mechanism is at work in *Xenopus* oocytes or zygotes, simply due to the dimensions and how yolk and other organelles are organized in these cells. However, these observations in *Xenopus* extracts reminded me very much to mammalian (mouse) oocytes. While quite some work has been done directly in mouse, also on the biophysical side (e.g. by the Verlhac/Terret laboratory), I am not aware of a study that would provide a similarly detailed mechanistic model, and comparably rigorous test of the model. Therefore, the present study will be inspirational/instructive for groups working on mammalian oocytes and early embryos. I also agree that addition of different cell cycle states was critical.

We thank the reviewer for their appreciation of our work.

In the current manuscript, the weakest link is that de-centering/symmetry breaking is major point even mentioned in the title, but in fact the mechanism of centering is what really is worked out in detail. The model for de-centering is rather coarse, and I would be happy to see more discussion on this even if it is a bit on the speculative side.

We have expanded the discussion of the symmetry breaking mechanism as requested, both in the Results section and in the Discussion section. The main point we want to emphasize is that the observed phenomenology, with the time and size-dependent localization pattern, is primarily dependent on the transient nature of the surface interaction, in conjunction with the continuous hydrodynamic centering force.

Reviewer 3:The only quantitative comparison between experiments and the model are the displacement vs. time curves for the aggregate recentering dynamics (e.g. Figure 4). It would nice to include an additional comparison. For example, the authors could include a plot of the recentering velocity vs displacement curves like in Figure 3G that compares experiments and simulations for the I-phase and M-phase extracts.

We thank the reviewer for their suggestion. We have added a supplementary figure with a plot of the recentering velocity vs displacement for the I-phase and M-phase extracts in experiment and simulation (Figure 4—figure supplement 2 in the revised manuscript).